# Tetrahedral triple-Q magnetic ordering and large spontaneous Hall conductivity in the metallic triangular antiferromagnet $Co_{1/3}TaS_2$

Pyeongjae Park [1,2], Woonghee Cho[1,2], Chaebin Kim [1,2], Yeochan An [1,2], Yoon-Gu Kang [3], Maxim Avdeev [4,5], Romain Sibille [6], Kazuki Iida [7], Ryoichi Kajimoto [8], Ki Hoon Lee[9], Woori Ju[10], En-Jin Cho[10], Han-Jin Noh[10], Myung Joon Han [3], Shang-Shun Zhang[11], Cristian D. Batista [12,13] & Je-Geun Park [1,2,14]

The triangular lattice antiferromagnet (TLAF) has been the standard paradigm of frustrated magnetism for several decades. The most common magnetic ordering in insulating TLAFs is the 120° structure. However, a new triple-**Q** chiral ordering can emerge in metallic TLAFs, representing the short wavelength limit of magnetic skyrmion crystals. We report the metallic TLAF $Co_{1/3}TaS_2$ as the first example of tetrahedral triple-**Q** magnetic ordering with the associated topological Hall effect (non-zero $\sigma_{xy}(\mathbf{H}=0)$). We also present a theoretical framework that describes the emergence of this magnetic ground state, which is further supported by the electronic structure measured by angle-resolved photoemission spectroscopy. Additionally, our measurements of the inelastic neutron scattering cross section are consistent with the calculated dynamical structure factor of the tetrahedral triple-**Q** state.

Discovering magnetic orderings with novel properties and functionalities is an important goal of condensed matter physics. While ferromagnets are still the best examples of functional magnetic materials due to their vast spectrum of technological applications, antiferromagnets are creating a paradigm shift for developing new spintronic components[1,2]. A more recent case is the discovery of Skyrmion crystals induced by a magnetic field in different classes of materials. These chiral states, which result from a superposition of three spirals with ordering wave vectors that differ by a ±120° rotation about a high-symmetry axis, can produce a substantial synthetic magnetic field that couples only to the orbital degrees of freedom of conduction electrons[3,4]. When conduction electrons propagate through a skyrmion spin texture, they exhibit a spontaneous Hall effect. The origin of this effect becomes transparent in the adiabatic limit. Due to the

[1]Center for Quantum Materials, Seoul National University, Seoul 08826, Republic of Korea. [2]Department of Physics & Astronomy, Seoul National University, Seoul 08826, Republic of Korea. [3]Department of Physics, KAIST, Daejeon 34141, Republic of Korea. [4]Australian Nuclear Science and Technology Organisation (ANSTO), New Illawarra Road, Lucas Heights, NSW 2234, Australia. [5]School of Chemistry, The University of Sydney, Sydney, NSW 2006, Australia. [6]Laboratory for Neutron Scattering and Imaging, Paul Scherrer Institut, 5232 Villigen, Switzerland. [7]Comprehensive Research Organization for Science and Society (CROSS), Tokai, Ibaraki 319-1106, Japan. [8]Materials and Life Science Division, J-PARC Center, Japan Atomic Energy Agency, Tokai, Ibaraki 319-1195, Japan. [9]Department of Physics, Incheon National University, Incheon 22012, Republic of Korea. [10]Department of Physics, Chonnam National University, Gwangju 61186, Republic of Korea. [11]School of Physics and Astronomy and William I. Fine Theoretical Physics Institute, University of Minnesota, Minneapolis, MN 55455, USA. [12]Department of Physics and Astronomy, The University of Tennessee, Knoxville, TN 37996, USA. [13]Quantum Condensed Matter Division and Shull-Wollan Center, Oak Ridge National Laboratory, Oak Ridge, TN 37831, USA. [14]Institute of Applied Physics, Seoul National University, Seoul 08826, Republic of Korea. ✉e-mail: cbatist2@utk.edu; jgpark10@snu.ac.kr

exchange interaction, the underlying local moment texture aligns the spin of a conduction electron that moves in a loop, inducing a Berry phase in its wavefunction equal to half of the solid angle spanned by the local moments enclosed by the loop. This phase is indistinguishable from the Aharonov-Bohm phase induced by a real magnetic flux, and each skyrmion generates a flux quantum in the corresponding magnetic unit because the spins span the full solid angle of the sphere ($4\pi$).

The triangular lattice Heisenberg model is a textbook example that can host diverse quantum states with small variations of short-range exchange interactions. The generic ground state for nearest-neighbor (NN) antiferromagnetic interactions is the three-sublattice 120° structure shown in Fig. 1a. This spiral structure is characterized by an ordering wave vector located at $\pm$K-points of the hexagonal Brillouin zone (Fig. 1d). Adding a relatively small second NN anti-ferromagnetic interaction gives rise to the two-sublattice collinear stripe spin configuration shown in Fig. 1b, whose ordering wave vector is one of the three M-points of the Brillouin zone (see Fig. 1e). Remarkably, for $S = \frac{1}{2}$, these two phases seem to be separated by a quantum spin liquid state[5-11] whose nature is not yet fully understood.

However, small effective four-spin interactions can induce a fundamentally different chiral antiferromagnetic order in triangular lattice antiferromagnets. This state is the triple-**Q** version of the stripe order, where the three different M-ordering wave vectors (see Fig. 1f) coexist in the same phase giving rise to a non-coplanar four-sublattice magnetic ordering (see Fig. 1c). The spins of each sublattice point along the all-in or all-out principal directions of a regular tetrahedron. Theoretical studies suggest that this state can appear naturally in metallic TLAFs, where effective four-spin interactions arise from the exchange interaction between conduction electrons and localized spin degrees of freedom[12,13]. This state was predicted to appear in the Mn monolayers on Cu(111) surfaces by density functional theory[14], and it was observed in the hcp Mn monolayers on Re(0001) using spin-polarized scanning tunneling microscopy[15,16]. However, it has not been reported yet in bulk systems.

The tetrahedral ordering is noteworthy for its topological nature, as it can be viewed as the short-wavelength limit of a magnetic skyrmion crystal[17]. The three spins of each triangular plaquette span one-quarter of the solid angle of a sphere, implying that each skyrmion (one flux quantum) is confined to four triangular plaquettes. As

illustrated in Fig. 1c, the two-dimensional (2D) magnetic unit cell of the tetrahedral ordering consists of eight triangular plaquettes, meaning there are two skyrmions per magnetic unit cell. In other words, the tetrahedral triple-**Q** ordering creates a very strong, effective magnetic field of one flux quantum divided by the area of 4 triangular plaquettes in the adiabatic limit. Notably, this ordering does not have any net spin magnetization. However, the emergent magnetic field couples to the orbital degrees of freedom of the conduction electrons, giving rise to a uniform orbital magnetization and a large topological Hall effect characterized by scalar spin chirality ($\chi_{ijk} = \langle \boldsymbol{S}_i \cdot \boldsymbol{S}_j \times \boldsymbol{S}_k \rangle$)[3,4]. Thus, this spin configuration is the simplest textbook example where non-trivial band topology is induced in the absence of relativistic spin-orbit coupling (the non-coplanar configuration generates an effective gauge field that couples to the orbital degrees of freedom of the conduction electrons). Moreover, the tetrahedral ordering can provide a potential route to realize the antiferromagnetic Chern insulator by properly adjusting the Fermi level of the system, as suggested in Refs. 12,13,18.

This work reports a four-sublattice tetrahedral triple-**Q** ordering as the only scenario known to the authors that is consistent with our data for the metallic triangular antiferromagnet $Co_{1/3}TaS_2$. Our key observations are the coexistence of long-range antiferromagnetic ordering with wave vector $\mathbf{q}_m = (1/2, 0, 0)$, a weak ferromagnetic moment ($M_z(\mathbf{H} = 0)$), and a non-zero AHE ($\sigma_{xy}(\mathbf{H} = 0)$) below $T_{N2}$, which rules out the possibility of single-**Q** and double-**Q** ordering[19-22]. Based on the crystalline and electronic structure of $Co_{1/3}TaS_2$, we also provide a theoretical conjecture about the origin of the observed ordering, consistent with our angle-resolved photoemission spectroscopy (ARPES) data. Moreover, the calculated low-energy magnon spectra of the tetrahedral ordering agree with the spectra measured by inelastic neutron scattering. Finally, we discuss the robustness of the tetrahedral ordering against an applied magnetic field in $Co_{1/3}TaS_2$.

## Results and discussion

$Co_{1/3}TaS_2$ is a Co-intercalated metal comprising triangular layers of magnetic $Co^{2+}$ ions (Fig. 1g). Previous studies on $Co_{1/3}TaS_2$ in the 1980s reported the bulk properties of a metallic antiferromagnet with $S = 3/2$ (a high-spin $d^7$ configuration of $Co^{2+}$)[23-25], including a neutron diffraction study that reported an ordering wave vector $\mathbf{q}_m = (1/3, 1/3, 0)$ characteristic of a 120° ordering[25]. More recently, an experimental study on single-crystal $Co_{1/3}TaS_2$ observed a significant

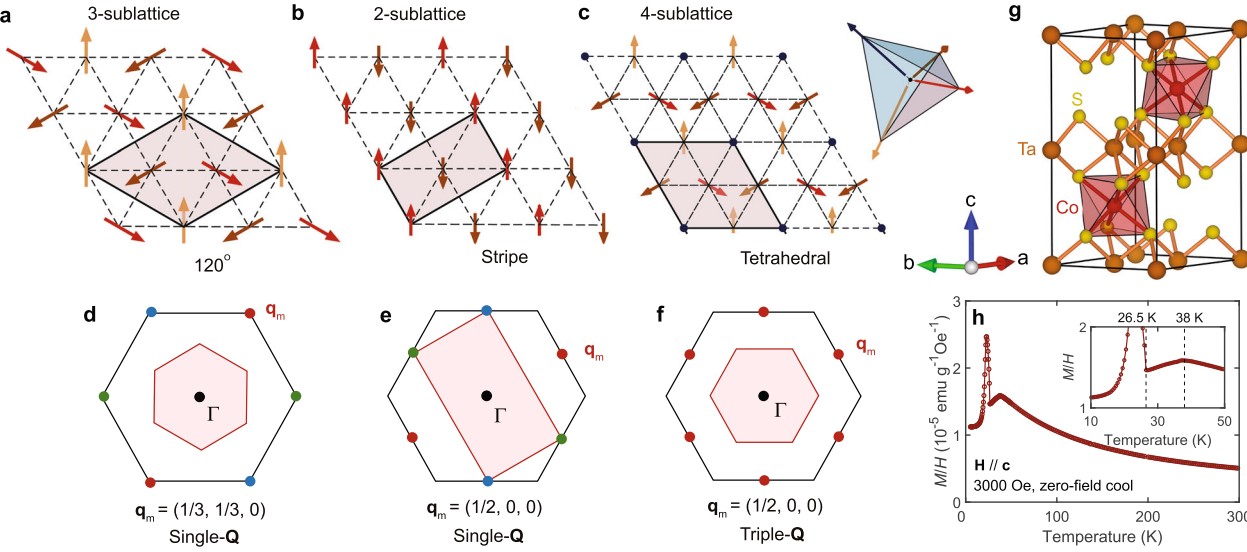

**Fig. 1 | The Tetrahedral triple-Q state and crystal structure of $Co_{1/3}TaS_2$.**
**a–c** Three fundamental antiferromagnetic orderings for a triangular lattice system. The red-shaded regions denote each magnetic unit cell. **d–f** Positions of the magnetic Bragg peaks (red circles) in momentum space generated by **a–c** A black (red)

hexagon corresponds to a crystallographic (magnetic) Brillouin zone. The green and blue circles in **d–e** denote the magnetic Bragg peaks from the other two magnetic domains. **g** A crystallographic unit cell of $Co_{1/3}TaS_2$. **h** The temperature-dependent magnetization of single-crystal $Co_{1/3}TaS_2$ with **H//c**.

anomalous Hall effect (AHE) comparable to those in ferromagnets below 26.5 K, which is the second transition temperature ($T_{N2}$) of the two antiferromagnetic phase transitions at $T_{N1} = 38$ K and $T_{N2} = 26.5$ K (see Fig. 1h)[26]. Based on the 120° ordering reported in ref. 25 and a symmetry argument, the authors of ref. 26 suggested that the observed AHE, $\sigma_{xy}(\mathbf{H} = 0) \neq 0$, can be attributed to a ferroic order of cluster toroidal dipole moments. However, our latest neutron scattering data reported in this work reveals an entirely different picture: $Co_{1/3}TaS_2$ has a magnetic structure with ordering wave vectors of the M-points ($\mathbf{q}_m = (1/2, 0, 0)$ and symmetry-related vectors) instead of $\mathbf{q}_m = (1/3, 1/3, 0)$. The most likely scenario for such distinct outcomes would be a difference in Co composition; while we confirmed that our experimental results are consistently observed in $Co_xTaS_2$ with $0.299(4) < x < 0.325(4)$, the reported composition value for ref. 25's sample is $x = 0.29$ (see ref. 24). Additional Co disorder, beyond variations in composition, could also be a contributing factor. However, the limited data available for the sample in ref. 25 prevents us from forming a definitive assessment in this regard (see Supplementary Notes). In any case, the theoretical analysis in ref. 26 based on ref. 25 is not valid for $\mathbf{q}_m = (1/2, 0, 0)$ since a different ordering wave vector leads to a qualitatively different scenario. This forced us to conduct a more extensive investigation and come up with a scenario consistent with our comprehensive experimental datasets.

Figure 2a, b show the neutron diffraction patterns of powder and single-crystal $Co_{1/3}TaS_2$ for $T < T_N$. Magnetic reflections appear at the M-points of the Brillouin zone for both $T < T_{N2}$ and $T_{N2} < T < T_{N1}$, implying that the ordering wave vector is $\mathbf{q}_m = (1/2, 0, 0)$ or its symmetrically equivalent wave vectors. It is worth noting that this observation is inconsistent with the previously reported wave vector $\mathbf{q}_m = (1/3, 1/3, 0)$[25]; i.e., $Co_{1/3}TaS_2$ does not possess a 120° magnetic ordering. The magnetic Bragg peaks at the three different M points connected by the three-fold rotation along the c-axis ($C_{3z}$) have equivalent intensities within the experimental error (Fig. 2c). This result suggests either single-$\mathbf{Q}$ or double-$\mathbf{Q}$ ordering with three equally weighted magnetic domains or triple-$\mathbf{Q}$ ordering.

First, we analyzed the neutron diffraction data based on group representation theory and Rietveld refinement, assuming a single-$\mathbf{Q}$ magnetic structure: $\mathbf{M}_i^\nu = \mathbf{\Delta}_\nu \cos(\mathbf{q}_m^\nu \bullet \mathbf{r}_i)$ with $\nu = 1, 2,$ and 3 (see Supplementary Text and Supplementary Tables 2 and 3). As a result, we found that the spin configurations for $T < T_{N2}$ and $T_{N2} < T < T_{N1}$ correspond to Fig. 2g, h, respectively, where each configuration belongs to the $\Gamma_2 + \Gamma_4$ ($\alpha \mathbf{V}_{22}^\nu + \beta \mathbf{V}_{41}^\nu$, see Supplementary Table 3) and $\Gamma_2$ ($\mathbf{V}_{22}^\nu$) representations—see Supplementary Notes. The refinement result yielded an ordered moment of $1.27(1)\mu_B/Co^{2+}$ at 3 K.

However, a single-$\mathbf{Q}$ spin configuration with $\mathbf{q}_m = (1/2, 0, 0)$ possesses time-reversal symmetry (TRS) combined with lattice translation ($\equiv \tau_{1a}T$, see Fig. 2g, h), which strictly forbids the finite $\sigma_{xy}(\mathbf{H} = 0)$ and $M_z(\mathbf{H} = 0)$ observed at $T < T_{N2}$ (Fig. 2e, f). Similarly, double-$\mathbf{Q}$ spin configurations in a triangular lattice are not compatible with finite $\sigma_{xy}(\mathbf{H} = 0)$ and $M_z(\mathbf{H} = 0)$ due to its residual symmetry relevant to chirality cancellation[19]. Triple-$\mathbf{Q}$ ordering is, therefore, the only possible scenario that can resolve this contradiction since it allows for finite $\sigma_{xy}(\mathbf{H} = 0)$ and $M_z(\mathbf{H} = 0)$ due to broken $\tau_{1a}T$ symmetry[19–21]. In general, determining whether the magnetic structure of a system is a single/double-$\mathbf{Q}$ phase with three magnetic domains or a triple-$\mathbf{Q}$ ordering requires advanced experiments. However, $\mathbf{q}_m = (1/2, 0, 0)$ is a special case where a triple-$\mathbf{Q}$ state can be easily distinguished from the other possibilities by probing non-zero TRS-odd quantities incompatible with the symmetry of single/double-$\mathbf{Q}$ structures.

The most symmetric triple-$\mathbf{Q}$ ordering that produces the same neutron diffraction pattern as that of Fig. 2h is illustrated in Fig. 2i. This is precisely the four-sublattice tetrahedral ordering shown in Fig. 1c, except that $Co_{1/3}TaS_2$ has an additional 3D structure with an AB stacking pattern. Such a triple-$\mathbf{Q}$ counterpart can be obtained through a linear combination of three symmetrically equivalent single-$\mathbf{Q}$ states ($\mathbf{M}_i^\nu = \mathbf{\Delta}_\nu \cos(\mathbf{q}_m^\nu \bullet \mathbf{r}_i)$ with $\nu = 1, 2, 3$) connected by the three-fold rotation about the $c$ axis[22] (see Supplementary Notes for more explanation). However, an arbitrary linear combination of these three components yields a triple-$\mathbf{Q}$ spin configuration with a site-dependent magnitude of ordered moments on the Co sites, i.e., $|\mathbf{M}_i^{tri}|$ depends on $i$ (see Supple-

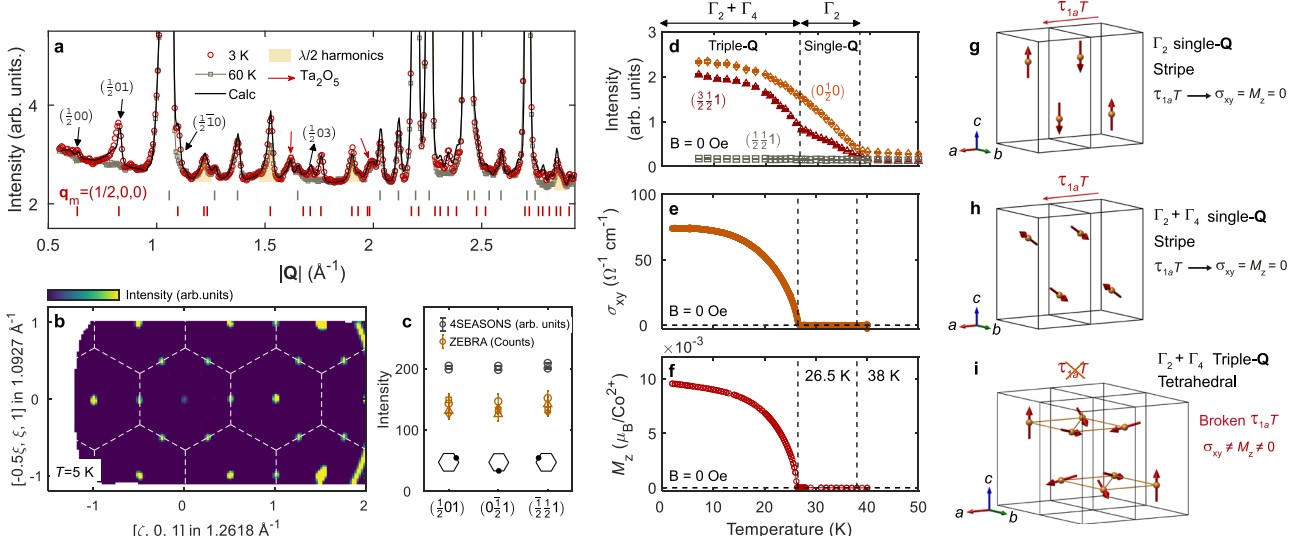

**Fig. 2 | Magnetic ground state of $Co_{1/3}TaS_2$ revealed by neutron diffraction.** **a** The powder neutron diffraction pattern of $Co_{1/3}TaS_2$, measured at 60 K (Gray squares) and 3 K (Red circles). A weak $\lambda/2$ signal is highlighted explicitly (see Methods). The solid black line is the diffraction pattern of $Co_{1/3}TaS_2$ simulated with the spin configuration shown in **h**, or equivalently **i**. The gray (red) vertical solid lines denote the position of nuclear (magnetic) reflections. The full diffraction data can be found in Supplementary Fig. 3. **b** The single-crystal neutron diffraction pattern at 5 K, demonstrating magnetic Bragg peaks located at the M points. **c** The intensities of the three magnetic Bragg peaks originating from three different ordering wave vectors. **d** The temperature-dependent intensities of some magnetic Bragg peaks in single-crystal $Co_{1/3}TaS_2$. **e, f** The temperature dependence of $\sigma_{xy}(\mathbf{H} = 0)$ and $M_z(\mathbf{H} = 0)$ in $Co_{1/3}TaS_2$, measured after field cooling under 5 T. **g, h** The refined magnetic structures for **g** 26.5 K < T < 38 K and **h** T < 26.5 K. **i** The triple-$\mathbf{Q}$ counterpart (tetrahedral) of the single-$\mathbf{Q}$ (stripe) ordering shown in **h**. Note that the spin configurations in **h** and **I** give the same powder diffraction pattern.

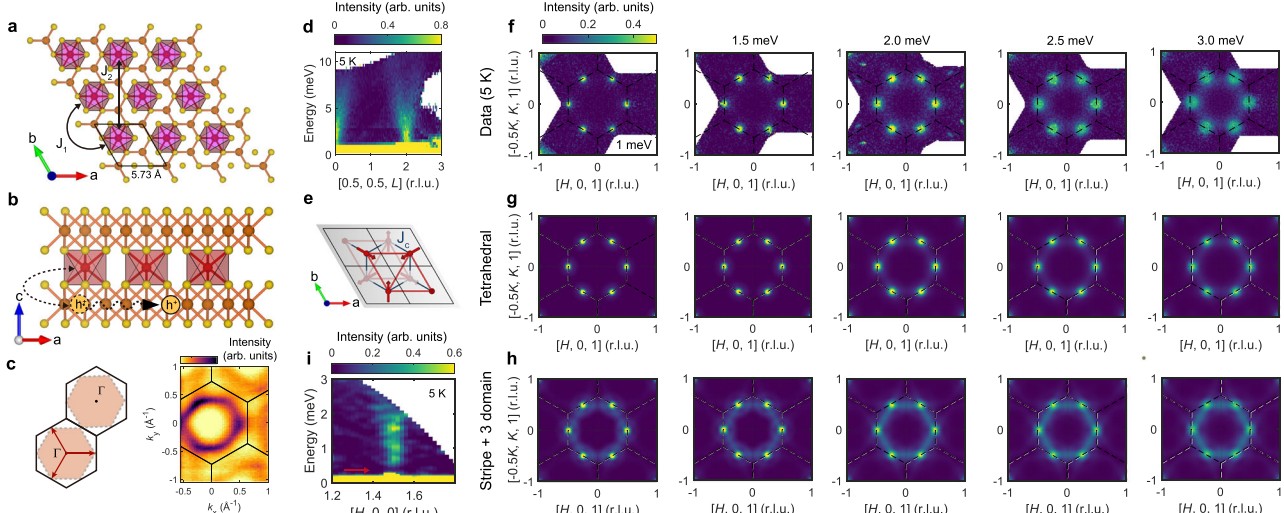

**Fig. 3 | Stabilization mechanism and dynamical properties of the tetrahedral order in Co$_{1/3}$TaS$_2$. a** The in-plane crystal structure of Co$_{1/3}$TaS$_2$ demonstrates isolated CoS$_6$ octahedrons (purple-colored) and a long NN Co-Co distance. **b** The exchange interaction between Co local moments and conduction electrons from TaS$_2$ layers leads to an effective RKKY interaction between the local moments. **c** The Fermi surface of a 2D TLAF with 3/4 filling (shaded hexagons) and the Fermi surface of Co$_{1/3}$TaS$_2$ measured by ARPES. **d** The magnon spectra of Co$_{1/3}$TaS$_2$ at 5 K along the (00 L) direction. $E_i$ = 7.9 and 14 meV data are plotted. **e** Antiferromagnetic NN interlayer coupling ($J_c$) of Co$_{1/3}$TaS$_2$, which is necessary for explaining the data in **d** and the refined spin configuration (Fig. 2g–i). **f** Const-$E$ cuts of the INS data measured at 5 K ($<T_{N2}$). An energy integration range for each plot is $\pm$ 0.2 meV. The $E$ = 1 and 1.5 meV (2.0-3.0 meV) plots are based on the $E_i$ = 5 (7.9) meV data. In addition to bright circular spots centered at six M points (=linear modes), a weak, diffuse ring-like scattering which we interpret as the quadratic mode predicted by spin-wave theory, appears for $E$ > 1.5 meV. **g** The calculated INS cross-section of the tetrahedral triple-**Q** ordering with $J_1S^2$ = 3.92 meV, $J_cS^2$ = 2.95 meV, $J_2/J_1$ = 0.19, and $K_{bq}/J_1$ = 0.02 (see Supplementary Materials). **h** The calculated INS cross-section of the single-**Q** ordering with three domains, using $J_1S^2$ = 3.92 meV, $J_cS^2$ = 2.95 meV, $J_2/J_1$ = 0.1, and $K_{bq}/J_1$ = 0. The line-shaped signal in **h** has a much higher intensity than in **f** or **g**. The simulations in **g**, **h** include resolution convolution (see Supplementary Fig. 9), and their momentum and energy integration range are the same as **f**. **i** Low-energy magnon spectra measured with $E_i$ = 3.5 meV at 5 K, showing the energy gap of the linear magnon mode.

mentary Fig. 7). A uniform $|\mathbf{M}_i^{tri}|$ is obtained only when the Fourier components $\boldsymbol{\Delta}_\nu$ of the three wave vectors $\mathbf{q}_m^\nu$ are orthogonal to each other[12], i.e., $\mathbf{M}_i^{tri} = \sum_{\nu=1,3} \boldsymbol{\Delta}_\nu \cos(\mathbf{q}_m^\nu \cdot \mathbf{r}_i)$ with $\boldsymbol{\Delta}_\nu \perp \boldsymbol{\Delta}_{\nu'}$ for $\nu \neq \nu'$. We note, however, that the magnitude of the ordered moments ($|\mathbf{M}_i^{tri}|$) does not have to be the same for quantum mechanical spins. Nevertheless, as explained in detail in the Supplementary Information, only non-coplanar triple-**Q** orderings corresponding to equilateral ($|\boldsymbol{\Delta}_\nu| = |\boldsymbol{\Delta}_{\nu'}|$, Fig. 2h) or non-equilateral ($|\boldsymbol{\Delta}_\nu| \| |\boldsymbol{\Delta}_{\nu'}|$) tetrahedral configurations are consistent with our Rietveld refinement of neutron diffraction data.

In the high-temperature ordered phase at $T_{N2} < T < T_{N1}$, the triple-**Q** ordering that yields the same neutron diffraction pattern as that of the single-**Q** ordering shown in Fig. 2g is collinear, giving rise to highly nonuniform $|\mathbf{M}_i^{tri}|$ (see Supplementary Fig. 7). Such a strong modulation of $|\mathbf{M}_i^{tri}|$ is unlikely for $S = 3/2$ moments weakly coupled to conduction electrons (see discussion below). In addition, the collinear triple-**Q** ordering allows for finite $M_z(\mathbf{H} = 0)$ and $\sigma_{xy}(\mathbf{H} = 0)$[19–22], while they are precisely zero within our measurement error for $T_{N2} < T < T_{N1}$. On the other hand, the single-**Q** ordering shown in Fig. 2g is more consistent with $M_z(\mathbf{H} = 0) = \sigma_{xy}(\mathbf{H} = 0) = 0$ in the temperature range $T_{N2} < T < T_{N1}$ due to its $\tau_{1a}T$ symmetry. Therefore, the combined neutron diffraction and anomalous transport data indicate that a transition from a collinear single-**Q** to non-coplanar triple-**Q** ordering occurs at $T_{N2}$. Interestingly, as we will see below, our theoretical analysis captures this two-step transition process.

We now examine the feasibility of the tetrahedral triple-**Q** ground state in Co$_{1/3}$TaS$_2$. Notably, in contrast to typical triple-**Q** orderings reported in other materials[27–29], this state emerges spontaneously in Co$_{1/3}$TaS$_2$ without requiring an external magnetic field. It was proposed on theoretical grounds that this state could arise in a 2D metallic TLAF formulated by the Kondo lattice model[12,13,18,30]:

$$H = -t \sum_{\langle i,j \rangle} c_{i\alpha}^\dagger c_{j\alpha} - J \sum_i \boldsymbol{S}_i \cdot c_{i\alpha}^\dagger \boldsymbol{\sigma}_{\alpha\beta} c_{i\beta}. \tag{1}$$

From a crystal structure perspective, Co$_{1/3}$TaS$_2$ is an ideal candidate to be described by this model. The nearest Co-Co distance (5.74 Å) is well above Hill's limit, and the CoS$_6$ octahedra are fully isolated ((Fig. 3a). This observation suggests that the Co 3d bands would retain their localized character, while itinerant electrons mainly arise from the Ta 5d bands. Our density functional theory (DFT) calculations confirm this picture and reveal that the density of states near the Fermi energy has a dominant Ta 5d orbital character (Supplementary Fig. 11). In this situation, a magnetic Co$^{2+}$ ion can interact with another Co$^{2+}$ ion only via the conduction electrons in the Ta 5d bands. Thus, in a first approximation, the Co$^{2+}$ 3d electrons can be treated as localized magnetic moments interacting via exchange with the Ta 5d itinerant electrons[23,25] (see Fig. 3b). Indeed, the Curie-Weiss behavior observed in Co$_{1/3}$TaS$_2$ provides additional support for this picture; the magnitude of the fitted effective magnetic moment indicates $S$-1.35, close to the single-ion limit of Co$^{2+}$ in the high spin $S = 3/2$ configuration[26]. However, our neutron diffraction measurement indicates a significant suppression of the ordered magnetic moment $S$~0.64 (assuming that a g-factor is 2). Similar reductions of the ordered moment have been reported in other metallic magnets comprising 3d transition metal elements, and they are generally attributed to a partial delocalization of the magnetic moments. In addition, interactions between Co local moments and itinerant electrons from Ta 5d band can lead to partial screening of the local moments. Another possible origin of the reduction of the ordered moment is quantum spin fluctuations arising from the frustrated nature of the effective spin-spin interactions.

Tetrahedral ordering can naturally emerge when the Fermi surface (FS) is three-quarters (3/4) filled[12,13,18,31] because the shape of the FS is a regular hexagon (for a tight-binding model with nearest-neighbor hopping), whose vertices touch the M-points of the first Brillouin zone (Fig. 3c). In this case, there are three nesting wave vectors connecting the edges of the regular hexagon and the van Hove singularities at

different M points, leading to magnetic susceptibility of the conduction electrons ($\chi(\mathbf{q}, E_F)$) that diverges as $\log^2 |\mathbf{q} - \mathbf{q}_m^\nu|$ at $\mathbf{q} \in$ M-points[30]. This naturally results in a magnetic state with ordering wave vectors corresponding to the three M points (i.e., triple-$\mathbf{Q}$). While perfect nesting conditions are not expected to hold for real materials, $\chi(\mathbf{q}, E_F)$ is still expected to have a global maximum at the three M points for Fermi surfaces with the above-mentioned hexagonal shape[18]. As shown on the right side of Fig. 3c, our ARPES measurements reveal that this is indeed the case of the FS of $Co_{1/3}TaS_2$, indicating that the filling fraction is close to 3/4 and that the effective interaction between the $Co^{2+}$ magnetic ions is mediated by the conduction electrons.

The above-described stabilization mechanism based on the shape of the Fermi surface suggests that the effective exchange interaction between the $Co^{2+}$ magnetic ions and the conduction electrons is weak compared to the Fermi energy. Under these conditions, it is possible to derive an effective RKKY spin Hamiltonian by applying degenerate second-order perturbation theory in $J/t$[30,31]. Since the RKKY model includes only bilinear spin-spin interactions, the single-$\mathbf{Q}$ (stripe) and triple-$\mathbf{Q}$ (tetrahedral) orderings remain degenerate in the classical limit. Moreover, by tuning the mutually orthogonal vector amplitudes $\mathbf{\Delta}_\nu$ while preserving the norm $|\mathbf{\Delta}_1|^2 + |\mathbf{\Delta}_2|^2 + |\mathbf{\Delta}_3|^2$, it is possible to continuously connect the single-$\mathbf{Q}$ (stripe) and triple-$\mathbf{Q}$ (tetrahedral) orderings via a continuous manifold of degenerate multi-$\mathbf{Q}$ orderings. The classical spin model's accidental ground state degeneracy leads to a gapless magnon mode with quadratic dispersion and linear Goldstone modes. The accidental degeneracy is broken by effective four-spin exchange interactions that gap out the quadratic magnon mode, which naturally arises from Eq. (1) when effective interactions beyond the RKKY level are taken into consideration[14,17,30,31]. The simplest example of a four-spin interaction favouring the triple-$\mathbf{Q}$ ordering is the bi-quadratic term $K_{bq}(\mathbf{S}_i \cdot \mathbf{S}_j)^2$ with $K_{bq} > 0$. This was explicitly demonstrated using classical Monte-Carlo simulations with a simplified phenomenological $J_1$-$J_2$-$J_c$-$K_{bq}$ model (see Fig. 3a and 3e), as shown in Supplementary Fig. 8. This model successfully captures the tetrahedral ground state and manifests a two-step transition process at $T_{N2}$ and $T_{N1}$ with an intermediate single-$\mathbf{Q}$ ordering, which is consistent with our conclusion based on experimental observations. The latter outcome can be attributed to thermal fluctuations favouring the collinear spin ordering[32,33]. In addition, previous DFT studies on $TM_{1/3}NbS_2$ (TM = Fe, Co, Ni), which is isostructural to $Co_{1/3}TaS_2$, also revealed that a non-coplanar state could be energetically more favorable than collinear and co-planar states[34].

Before analyzing the magnon spectra in detail, it is worth considering the interlayer network of $Co_{1/3}TaS_2$ with AB stacking. Along with the refined magnetic structure (Fig. 2g–i), the steep magnon dispersion shown in Fig. 3d indicates non-negligible antiferromagnetic NN interlayer exchange: $J_c S^2 \sim 2.95$ meV (see Fig. 3e). However, the finite value of $J_c$ does not change the competition between stripe and tetrahedral orderings analyzed in the 2D limit. More importantly, the antiferromagnetic exchange $J_c$ forces the tetrahedral spin configuration of the B layer to be the same as that of the A layer (see Supplementary Fig. 8a). Therefore, all triangular layers have the same sign of the scalar chirality $\chi_{ijk}$ (or skyrmion charge), resulting in the realization of 3D ferro-chiral ordering. In this 3D structure, each magnetic unit cell of $Co_{1/3}TaS_2$ includes four skyrmions.

The low-energy magnon spectra (<3 meV) of $Co_{1/3}TaS_2$ measured by INS are presented in Fig. 3f. In addition to the linear (Goldstone) magnon modes, which appear as bright circular signals centered at the M points, a ring-like hexagonal signal connecting six M points was additionally observed with weaker intensity. This can be interpreted as the trace of the quadratic modes since they should be present together with linear modes at low energy (see Supplementary Fig. 9). Since the signal is only present for $E > 1.5$ meV, we infer that the quadratic mode is slightly gapped. We used linear spin-wave theory to compare the measured INS spectra with the theoretical spectra of both single-$\mathbf{Q}$

and triple-$\mathbf{Q}$ orderings[30]. Despite the simplicity of the $J_1$-$J_2$-$J_c$-$K_{bq}$ model, the calculated magnon spectra of the tetrahedral ordering (Fig. 3g) successfully describe the measured INS spectra. The magnon spectra of the stripe order are also presented in Fig. 3h for comparison. The intensity of the quadratic magnon mode is much stronger than that of the triple-$\mathbf{Q}$ spectra, in apparent disagreement with our INS data. A complete comparison between our data and the two calculations is shown in Supplementary Fig. 10. Additionally, the linear magnon modes of $Co_{1/3}TaS_2$ are also slightly gapped (~0.5 meV, see Fig. 3i). This feature can be explained for tetrahedral ordering only by considering both exchange anisotropy and higher order corrections in the $1/S$ expansion (see Supplementary Notes).

Finally, we discuss the behavior of the tetrahedral ordering in $Co_{1/3}TaS_2$ in response to out-of-plane and in-plane magnetic fields. Figure 4c illustrates the field dependence of the measured anomalous Hall conductivity ($\sigma_{xy}^{AHE}$) and $M_z$ for $\mathbf{H} // \mathbf{c}$, showing evident hysteresis with a sign change at $\pm H_{c1}$. Indeed, as already discussed in ref. 26, $M_z$ cannot characterize the observed $\sigma_{xy}^{AHE}$. Instead, as explained in the introduction, it is $\chi_{ijk}$ that characterizes $\sigma_{xy}^{AHE}$ for the tetrahedral ordering. Therefore, based on our triple-$\mathbf{Q}$ scenario (see Fig. 4a, b), the sign change at $\pm H_{c1}$ can be interpreted as the transition between tetrahedral orderings with positive and negative values of the scalar chirality $\chi_{ijk}$. In addition, the coexistence of weak ferromagnetic moment and large $\sigma_{xy}(\mathbf{H} = 0)$ is expected because, as we explained before, the real-space Berry curvature of the tetrahedral triple-$\mathbf{Q}$ ordering generates both the orbital ferromagnetic moment (of conduction electrons) and spontaneous Hall conductivity. However, the experimental techniques used in this study cannot discriminate between the spin and orbital contributions to $M_z(\mathbf{H} = 0)$. While it will be interesting to identify the nature of this weak ferromagnetic moment, this is left for future studies.

We also compared the field dependence of the measured $\sigma_{xy}^{AHE}$ and $\chi_{ijk}$ calculated from our $M_z(\mathbf{H})$ data based on the canting expected in the tetrahedral order of $Co_{1/3}TaS_2$ (blue and orange arrows in Fig. 4). As shown in Fig. 4d, $\sigma_{xy}^{AHE}$ decreases slightly in response to both positive and negative magnetic fields, consistent with the calculated $\chi_{ijk}$ of the tetrahedral ordering with mild canting. However, the model cannot capture the sudden decrease of $\sigma_{xy}^{AHE}$ due to a meta-magnetic transition at $\pm H_{c2}$, indicating that this transition changes the tetrahedral spin configuration. Additional neutron diffraction is required to identify the new ordering for $|\mathbf{H}| > H_{c2}$.

The effect of an in-plane magnetic field was investigated using single-crystal neutron diffraction. Figure 4e shows field-dependent ($\mathbf{H} // \mathbf{q}_m^1 = (1/2, 0, 0)$) intensities of the three magnetic Bragg peaks, each originating from three different $\mathbf{q}_m^\nu$ of the tetrahedral ordering. Interestingly, the equal intensity of the three peaks remains almost unchanged by a magnetic field up to 10 T, indicating robustness of the tetrahedral ordering against an in-plane magnetic field. This should be contrasted with triple-$\mathbf{Q}$ states found in other materials, which are induced by a finite magnetic field and occupy narrow regions of the phase diagram due to the ferromagnetic nature of the dominant exchange interaction[27–29].

In summary, we have reported a tetrahedral triple-$\mathbf{Q}$ ordering in $Co_{1/3}TaS_2$, as the only magnetic ground state compatible with our bulk properties (non-zero $\sigma_{xy}(\mathbf{H} = 0)$ and $M_z(\mathbf{H} = 0)$) and neutron diffraction data ($\mathbf{q}_m = (1/2, 0, 0)$ or its symmetry-equivalent pairs). Moreover, we provide a complete theoretical picture of how this exotic phase can be stabilized in the triangular metallic magnet $Co_{1/3}TaS_2$, which is further corroborated by our measurements of electronic structure and long-wavelength magnetic excitations using ARPES and INS. We further investigated the effect of external magnetic fields on this triple-$\mathbf{Q}$ ordering, which demonstrates its resilience against the fields. Our study opens avenues for exploring chiral magnetic orderings with the potential for spontaneous integer quantum Hall effect[30].

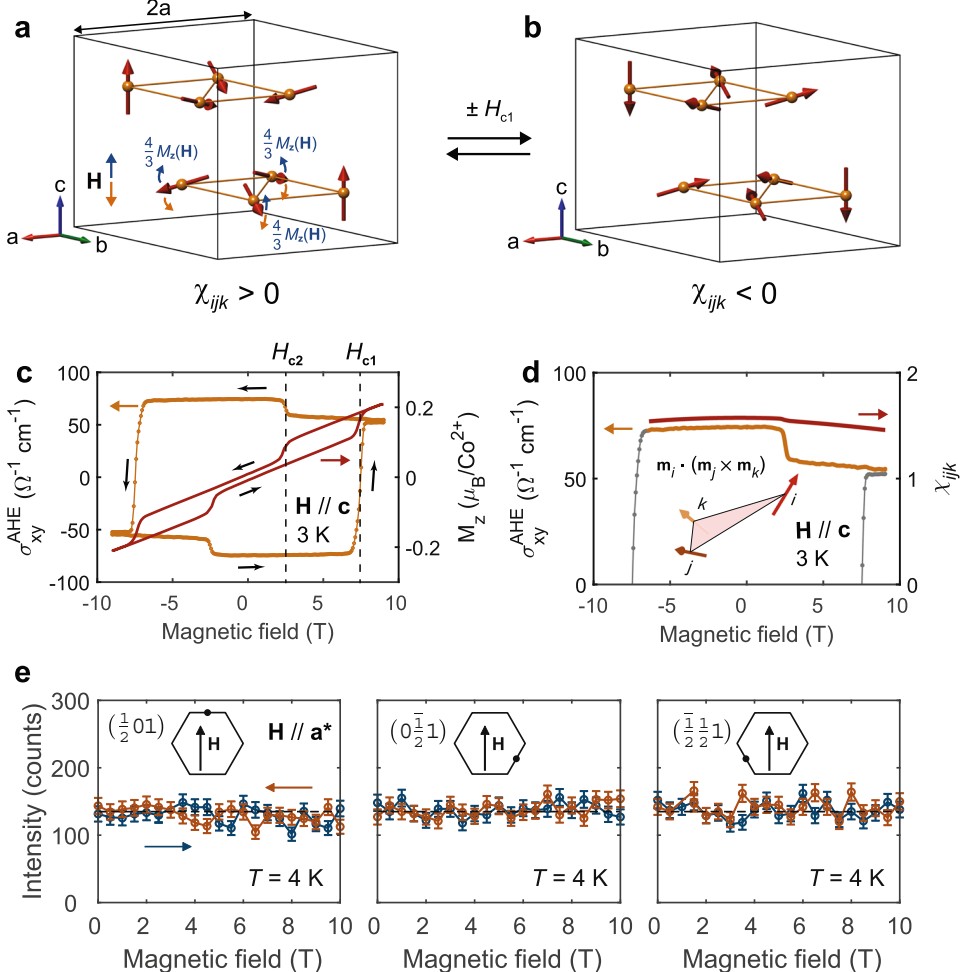

**Fig. 4 | The effect of the out-of-plane and in-plane magnetic field on the tetrahedral triple-Q ordering in $Co_{1/3}TaS_2$. a, b** A time-reversal pair of the tetrahedral spin configuration, having $\chi_{ijk}$ opposite to each other. The blue and orange arrows in **a** depict the generic canting of the tetrahedral ordering in $Co_{1/3}TaS_2$ by an out-of-plane magnetic field. **c** Comparison between the measured $\sigma_{xy}^{AHE}$ (orange) and $M_z$ (red) under the out-of-plane field at 3 K. **d** Comparison between the measured $\sigma_{xy}^{AHE}$ (orange) at 3 K and $\chi_{ijk}$ (red) calculated from the $M_z$ data in **c**. **e** Intensities of the three magnetic Bragg peaks in Fig. 2c under an external magnetic field along the **a\*** direction. Error bars in **e** represent the standard deviation of the measured intensity.

Note added: After the submission of this work, we came across another neutron diffraction work on $Co_{1/3}TaS_2$ published recently[35]. While they have drawn the same conclusion of magnetic structure as our tetrahedral triple-Q magnetic ordering, our study further delves into the microscopic aspects of this magnetic ground state, specifically in relation to electronic structure and the properties of a geometrically frustrated triangular lattice. We also substantiate our findings through the ARPES and INS measurements.

## Methods

### Sample preparation and structure characterization

Polycrystalline $Co_{1/3}TaS_2$ was synthesized by the solid-state reaction. A well-ground mixture of Co (Alfa Aesar, >99.99%), Ta (Sigma Aldrich, >99.99%), and S (Sigma Aldrich, >99.999%) was sealed in an evacuated quartz ampoule and then sintered at 900 °C for 10 days. A molar ratio of the three raw materials was $\alpha$:3:6 with $1.05 < \alpha < 1.1$, which is necessary to obtain the resultant stoichiometric ratio of Co close to 1/3. Single-crystal $Co_{1/3}TaS_2$ was grown by the chemical vapor transport method with an $I_2$ transport agent (4.5 mg $I_2/cm^3$). The pre-reacted polycrystalline precursor and $I_2$ were placed in an evacuated quartz tube and then were heated in a two-zone furnace with a temperature gradient from 940 to 860 °C for 10-14 days.

We measured the powder X-ray diffraction (XRD) pattern of $Co_{1/3}TaS_2$ using a high-resolution (Smartlab, Rigaku Japan) diffractometer, which confirmed the desired crystal structure without any noticeable disorder[26]. In particular, the intercalation profile of Co atoms was carefully checked by the (10 $L$) superlattice peak pattern in the low-$2\theta$ region; see ref. 26. The powder neutron diffraction experiment further corroborated this result (see the relevant subsection below). Finally, single crystals were examined by XRD (XtaLAB PRO, Cu $K_\alpha$, Rigaku Japan) and Raman spectroscopy (XperRam Compact, Nanobase Korea), confirming the high quality of our crystals, e.g., the sharp Raman peak at 137 $cm^{-1}$ (ref. 26).

The composition $x$ of $Co_xTaS_2$ single-crystal was confirmed primarily by energy-dispersive X-ray (EDX) spectroscopy (Quantax 100, Bruker USA & EM-30, Coxem Korea). We measured 12 square areas of 300μm × 300μm wide for every single piece, yielding homogeneous $x$ centered at ~0.320 with a standard deviation of ~0.004. The homogeneity of $x$ was further verified by the spatial profile (~1.5 μm resolution) of the EDX spectra, which is very uniform, as shown in Supplementary Fig. 1. The obtained $x$ from EDX was again cross-checked by the composition measured by inductively coupled plasma (ICP) spectroscopy (OPTIMA 8300, Perkin-Elmer USA), which is almost the same within a measurement error bar[26].

## Bulk property measurements

We measured the magnetic properties of $Co_{1/3}TaS_2$ using MPMS-XL5 and PPMS-14 with the VSM option (Quantum Design USA). To measure the spontaneous magnetic moment (Fig. 3d), a sample was field-cooled under 5 T and then measured without a magnetic field. Transport properties of $Co_{1/3}TaS_2$ were measured by using four systems: our home-built set-up, PPMS-9 (Quantum Design, USA), PPMS-14 (Quantum Design, USA), and CFMS-9T (Cryogenic Ltd, UK). To observe the temperature dependence of the anomalous Hall effect (Fig. 3d), we field-cooled the sample under ±9 T and then measured the Hall voltage without a magnetic field. The measured Hall voltage was anti-symmetrized to remove any longitudinal components. Hall conductivity ($\sigma_{xy}$) was derived using the following formula:

$$\sigma_{xy} = \frac{\rho_{xy}}{\rho_{xx}{}^2 + \rho_{xy}{}^2}. \qquad (2)$$

Anomalous Hall conductivity $\sigma_{xy}^{AHE}$ was derived by first subtracting a normal Hall effect from measured $\rho_{xy}$ and then using Eq. 2.

## Powder neutron diffraction

We carried out powder neutron diffraction experiments of $Co_{1/3}TaS_2$ using the ECHIDNA high-resolution powder diffractometer ($\lambda = 2.4395$ Å) at ANSTO, Australia. To acquire clear magnetic signals of $Co_{1/3}TaS_2$, which are weak due to the small content of Co and its small ordered moment, we used 20 g of powder $Co_{1/3}TaS_2$. The quality of the sample was checked before the diffraction experiment by measuring its magnetic susceptibility and high-resolution powder XRD. The neutron beam at ECHIDNA contains weak $\lambda/2$ harmonics (-0.3%), yielding additional nuclear Bragg peaks (Supplementary Fig. 3). We also performed Rietveld refinement and magnetic symmetry analysis using Fullprof software[36]. The results are summarized in Supplementary Tables 1–3 and Supplementary Figs. 2–4.

## Single-crystal neutron diffraction

We carried out single-crystal neutron diffraction under a magnetic field using ZEBRA thermal neutron diffractometer at the Swiss spallation neutron source SINQ. We used one single-crystal piece (-16 mg) for the experiment, which was aligned in a 10 T vertical magnet (Oxford instruments) with (HHL) horizontal (Supplementary Fig. 5a). The beam of thermal neutrons was monochromatic using the (220) reflection of germanium crystals, yielding a neutron wavelength of $\lambda = 1.383$ Å with >1% of $\lambda/2$ contamination. Bragg intensities as a function of temperature and magnetic field were measured using a single $^3$He-tube detector in front of which slits were appropriately adjusted.

## Angle-resolved photoemission spectroscopy (ARPES) measurements

The ARPES measurements were performed at the 4A1 beamline of the Pohang Light Source with a Scienta R4000 spectrometer[37]. Single crystalline samples with a large AHE were introduced into an ultra-high vacuum chamber and were cleaved in situ by a top post method at the sample temperature of -20 K under the chamber pressure of $\sim 5.0 \times 10^{-11}$ Torr. A liquid helium cryostat maintained the low sample temperature. The photon energy was set to 90 eV to obtain a high photoelectron intensity for the states of Co 3d characters. The total energy resolution was -30 meV, and the momentum resolution was -0.025 A$^{-1}$ in the measurements.

## Single-crystal inelastic neutron scattering

We performed single-crystal inelastic neutron scattering of $Co_{1/3}TaS_2$ using the 4SEASONS time-of-flight spectrometer at J-PARC, Japan[38]. For the experiment, we used nearly 60 pieces of the single-crystal with a total mass of 2.2 g, which were co-aligned on the Al sample holder with an overall mosaicity of -1.5° (Supplementary Fig. 5b). We mounted the sample with the geometry of the (H0L) plane horizontal and performed sample rotation during the measurement. The data were collected with multiple incident neutron energies (3.5, 5.0, 7.9, 14.0, and 31.6 meV) and the Fermi chopper frequency of 150 Hz using the repetition-rate-multiplication technique[39]. We used the Utsusemi[40] and Horace[41] software for the data analysis. Based on the crystalline symmetry of $Co_{1/3}TaS_2$, the data were symmetrized into the irreducible Brillouin zone to enhance statistics.

## Classical Monte-Carlo simulations

To investigate the magnetic phase diagram of our spin model, we performed a classical Monte-Carlo (MC) simulation combined with simulated annealing. We used Langevin dynamics[42] for the sampling method of a spin system. To search for a zero-temperature ground state, a large spin system consisting of $30 \times 30 \times 4$ unit cells (7200 spins) with periodic boundary conditions was slowly cooled down from 150 K to 0.004 K. The thermal equilibrium was reached at each temperature by evolving the system through 5000 Langevin time steps, with the length of each time-step defined as $dt = 0.02/(J_1 S^2)$. Finally, we adopted the final spin configuration at 0.004 K as the magnetic ground state, which was further confirmed by checking whether the same result was reproduced in the second trial.

For finite-temperature phase diagrams, we used a $30 \times 30 \times 6$ supercell (10,800 spins). After waiting for 600-2000 Langevin time steps for equilibration, 600,000-2,000,000 Langevin time steps were used for the sampling. Since thermalization and decorrelation time strongly depend on the temperature, the length of equilibration and sampling time steps were set differently for different temperatures. From this result, we calculated heat capacity ($C_V$), staggered magnetization ($M_{stagg}$), and total scalar spin chirality ($\chi_{ijk}$) using the following equations:

$$C_V = \frac{\langle \epsilon^2 \rangle - \langle \epsilon \rangle^2}{k_B T^2}, \qquad (3)$$

$$M_{stagg} = \frac{g\mu_B}{N} \sum_i (-1)^{2\pi(\mathbf{q}_m^\nu \cdot \mathbf{r}_i)} \langle \mathbf{S}_i \rangle, \qquad (4)$$

$$\chi_{ijk} = \frac{1}{S^3} \frac{\sum_\Delta \langle \mathbf{S}_{\Delta 1}(\mathbf{S}_{\Delta 2} \mathbf{S}_{\Delta 3}) \rangle}{N_t}, \qquad (5)$$

where $\langle A \rangle$ is an ensemble average of A estimated by the sampling, $\epsilon$ is the total energy per $Co^{2+}$ ion, $g$ is the Lande $g$-factor, $\mathbf{q}_m^\nu$ ($\nu = 1,2,3$) is the ordering wave vector of two-sublattice stripe order (see Supplementary Note), $\Delta$ is the index for a single triangular plaquette on a Co triangular lattice consisting of three sites ($\Delta_1$, $\Delta_2$, $\Delta_3$), and $N$ ($N_t$) is the total number of $Co^{2+}$ ions (triangular plaquettes). When calculating $M_{stagg}$, one should first identify which ordering wave vector the spin system chose among the three $\mathbf{q}_m^\nu$ ($\nu = 1,2,3$), and then use proper $\mathbf{q}_m^\nu$ to calculate $M_{stagg}$. For all the simulations, we used $S = 3/2$. The results of the simulations are shown in Supplementary Fig. 8.

## Spin-wave calculations

We calculated the magnon dispersion and INS cross-section of $Co_{1/3}TaS_2$ using linear spin-wave theory. For this calculation, we used the SpinW library[43]. Since we did not align positive and negative $\chi_{ijk}$ domains of the tetrahedral ordering in our INS experiments, we averaged the INS cross-section of both cases.

## DFT calculations

We performed first-principles calculations using 'Vienna ab initio simulation package (VASP)'[44-46] based on projector augmented wave (PAW) potential[47] and within Perdew-Burke-Ernzerhof (PBE) type of

GGA functional[48] (see Supplementary Fig. 11). DFT + $U$ method[49,50] was adopted to take into account localized Co-3$d$ orbitals properly, where $U = 4.1$ eV and $J_{Hund} = 0.8$ eV were used as obtained by the constrained RPA for CoO and Co[51]. For the $2 \times 2$ magnetic unit cell, we used the Γ-centered $6 \times 6 \times 6$ $k$-grid. We obtained and used the optimized crystal structure with the force criterion for the relaxation fixed at 1 meV/Å. The plane-wave energy cutoff was set to 500 eV.

## Data availability
The authors declare that data supporting the findings of this study are available within the paper and the Supplementary Information. Further datasets are available from the corresponding author upon request.

## Code availability
Custom codes used in this article are available from the corresponding author upon request.

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

## Acknowledgements

We acknowledge S. H. Lee, S. S. Lee, Y. Noda, and M. Mostovoy for their helpful discussions and M. Kenzelmann for his help with the experiments at SINQ. The Samsung Science & Technology Foundation supported this work (Grant No. SSTF-BA2101-05). The neutron scattering experiment at the Japan Proton Accelerator Research Complex (J-PARC) was performed under the user program (Proposal No. 2021B0049). One of the authors (J.-G.P.) is partly funded by the Leading Researcher Program of the National Research Foundation of Korea (Grant No. 2020R1A3B2079375). This work is based on experiments performed at the Swiss spallation neutron source SINQ, Paul Scherrer Institute, Villigen, Switzerland. C.D.B. acknowledges support from the U.S. Department of Energy, Office of Science, Office of Basic Energy Sciences, under Award No. DE-SC0022311.

## Author contributions

J.-G.P. initiated and supervised the project. P.P. synthesized the polycrystalline and single-crystal samples. P.P. performed all the bulk characterizations. M.A. carried out the powder neutron diffraction experiment. P.P. and R.S. performed the single-crystal neutron diffraction experiment at ZEBRA. P.P. analyzed the neutron diffraction data together with M.A.. P.P., C.K., Y.A., K.I., and R.K. conducted the single-crystal inelastic neutron scattering experiment at 4SEASONS. W.J., E.-J.C., and H.-J.N. conducted the ARPES experiment. Y.-G.K. and M.J.H. performed the DFT calculations. P.P., W.H.C., S.-S.Z., and C.D.B. conducted spin model calculations. P.P., W.H.C., S.-S.Z., K.H.L., Y.-G.K., M.J.H, H.-J.N., C.D.B., and J.-G.P. contributed to the theoretical analysis and discussion. P.P., C.D.B., and J.-G.P. wrote the manuscript with contributions from all authors.

## Competing interests

The authors declare no competing interests.
