## [Peer Review File · Nature Communications]

Reviewers' Comments:

Reviewer #1:

Remarks to the Author:

I have read carefully the revised draft of the paper and replies to my previous questions. Overall, I think that the paper has improved considerably and the authors have addressed most of my concerns. In principle, I recommend this manuscript for publication after the authors have answered the following questions:

1. The explanation of the ARPES data on the origin of the M-point triple-Q structure is problematic. In the Fermi surface nesting picture, the nesting susceptibility is proportional to the density of states at a certain nesting wavevector. In the case of hexagonal Fermi surfaces connecting the M-points, the nesting susceptibility should reach its maximum at the wavevectors connecting the edges of the FS hexagons, i.e., $(1/4 \ 1/4)$ and equivalent positions instead of $(1/2 \ 0)$ connecting the vertices, and the nesting susceptibility at the M point $(1/2 \ 0)$ should be at a saddle point instead of maximum. The authors should explain if the magnetic order is indeed driven by FS nesting, why is the magnetic order at $(1/2 \ 0)$ instead of $(1/4 \ 1/4)$.

2. What is the difference between the triple-Q and the stripe structure in terms of magnon dispersion? In principle, there are two differences. First, the stripe phase has two atoms per magnetic unit cell while the triple-Q phase has four, therefore the number of spin wave branches should be different, and the magnon dispersion at around Gamma and M should be identical for the triple-Q state. Second, the non-collinear spin structures will have more magnon-magnon coupling compared with the collinear one. Are the authors able to observe any of these effects in their INS spectra?

3. Have the authors compared the J1-J2-bq exchange model with J1-J2-J3 model? I understand the authors' argument that the J3 should be small even for the RKKY interactions, but it would be interesting to see the effect of J3 on the magnetic structure because it connects the sublattices with the same spin direction for the triple-Q structure, and a ferromagnetic J3 will favor the triple-Q structure over the stripe one.

Reviewer #2:

Remarks to the Author:

Even though the revised manuscript is markedly different from the original submission, it suffers from many of the same problems that I noted in my first review. The most important one is the misrepresentation of the authors' interpretations of the experimental results presented in the manuscript as actual experimental "discoveries". In my first review I noted that no conclusive evidence was presented in the manuscript that would support the main claim that a triple-Q ordering was actually observed. In the revised manuscript, authors present more data that agrees with such an interpretation, but this still does not amount to "discovery" of "Tetrahedral triple-Q ordering ...", which is the title claim of the manuscript. I noted that tendency to this type of grandiose thinking has already created problem for the authors' scientific credibility, where they have now "unobserved" the title claim of their previous work, "Field-tunable toroidal moment and anomalous Hall effect in non-collinear antiferromagnetic Weyl semimetal $\text{Co}_{1/3}\text{TaS}_2$ ". Nevertheless, in the present version of the manuscript authors keep confusing their interpretations, surmises, and assumptions with actual observations and experimental discoveries, reporting them as such.

Having said all of the above, I nevertheless am of an opinion that manuscript presents large amount of interesting high-quality data combining ARPES, neutron scattering, and bulk transport and magnetization measurement in a synergistic and very complementary fashion, and these results are of sufficiently high quality and impact to be publishable in Nature Communications. The results also lead to appealing interpretations, which, however (unfortunately), are not presented in a scientifically consistent and ethical manner. Hence, in order to be reconsidered for publication the manuscript needs to be significantly revised, including its title, abstract, and summary paragraph. Specifically, in the revised manuscript authors need to avoid grandiose claims where their hypotheses, surmises, opinions, and interpretations are presented as "observations" and

even "discoveries". Presently, such passages are encountered in every other paragraph of the manuscript. Below, I append a long list of specific comments on the issues which caught my attention.

Firstly, title needs to be changed to something like "large AHE in metallic antiferromagnet ...", or anything else, which however would faithfully describe experimental findings without misrepresenting an interpretation for an observation.

>30(Abstract) "The triangular lattice antiferromagnet (TLAF) has been the standard paradigm ..."
>49 "The triangular lattice Heisenberg model is a textbook example that can host a diverse ..."
>220 "However, the finite value of J_c does not change the qualitative features of the 2D stripe and tetrahedral orderings."

The finite (and large) J_c dispossesses the system of many peculiarities that a 2D triangular lattice antiferromagnet has, and this needs to be clearly explained. Stacked triangular-lattice systems with antiferromagnetic inter-layer coupling are, in many respects, fundamentally different from peculiar 2D triangular-lattice antiferromagnet that "has been the standard paradigm" and a "a textbook example" (3D-dimensionality strongly reduces quantum fluctuations, wave vector $Q = (1/3, 1/3, 1/2)$ is different from $2Q = (1/3, 1/3, 0)$, etc).

>40 "Discovering magnetic orderings with novel properties and functionalities is one of the main goals of condensed matter physics"

Magnetic orderings, even with novel properties and functionalities, is a narrow subfield of condensed matter physics, so the above claim is an example of grandiose thinking of which there are plenty in the manuscript.

>46-47 "can produce a huge synthetic magnetic field that couples only to the orbital degrees of freedom of conduction electrons."

The mechanism (of Coulomb origin) should be clearly explained. Namely, that in the presence of exchange interaction with conduction electrons these magnetic structures can induce such and such effects on conduction electron motion.

>50-51 "For any spin value, the generic ground state for nearest-neighbour (NN) antiferromagnetic interactions is the three-sublattice 120° structure shown in Fig. 1a "
"For any spin value" is redundant and of questionable validity. It should also be specified to which interactions this refers. In general case, spin-spirals are possible in row models and perhaps in models with general long-range interactions on triangular lattice.

>63-64 The spins of each sublattice point along the all-in or all-out principal directions of a regular tetrahedron.

Co appears to be in an octahedral environment, and in the case of an ideal octahedron has 3 symmetry-equivalent axes that could define moment (spin) direction. The fourth "tetrahedral" axis is not consistent with crystallographic environment and does not look very natural. Partially disordered phases have been considered and also observed experimentally on triangular lattices. It appears from what follows that the observed magnetic Bragg peaks can be described by a partially-disordered structure where the "fourth" spin pointing along a "crystallographically unnatural" direction is simply not ordered. There is no explanation why this possibility is discarded (see also related comments below).

>87-88 "This work reports the four-sublattice tetrahedral triple-Q ordering as the only viable scenario for the metallic triangular antiferromagnet $\text{Co}_{1/3}\text{TaS}_2$."

Specialize "only viable scenario" to "only viable scenario known to the present authors". The difference is same as that between "the best food" and "the best food we tasted". The four-sublattice structure with one sublattice disordered still triple-Q. Is it at all important how many Q's are needed to describe it? Are there other partially ordered magnetic structures that would be consistent with the observed set of magnetic Bragg peaks?

>82-83 "where non-trivial band topology is induced in the absence of relativistic spin-orbit coupling (the noncoplanar configuration generates the large spin-orbit coupling)."

The spin-orbit interaction nomenclature refers to a relativistic effect (see eg H. A. Bethe, R.

Jackiw, Intermediate Quantum Mechanics, Addison-Wesley (1997), pp. 152-178). The effects of exchange coupling between the ordered moments and conduction electrons are Coulomb effects, not spin-orbit interaction effects, and should be called what they are. This issue of unorthodox notations that are difficult to understand was a major unclarity in the previous manuscript, which I commented upon, and it still persists here. Authors might say that exchange effects can "transfer" spin chirality of magnetic structure to the orbital motion of conduction electrons leading to Hall effect etc, but not that they generate "spin-orbit coupling", which is conventionally understood as an atomic effect (see reference above).

>100-101 "previous neutron diffraction study reported the ordering wave vector $q_m = (1/3, 1/3, 0)$ characteristic of 120° ordering. In contrast, the latest experimental study of the single-crystal $\text{Co}_{1/3}\text{TaS}_2$ reported two antiferromagnetic phase transitions at $T_{N1} = 38$ K and $T_{N2} = 26.5$ K (Fig. 1h), as well as a significant anomalous Hall effect (AHE) comparable to that of ferromagnets below T_{N2} ."

What is "in contrast" here? Triangular ordering more often than not comes with two Neel temperatures, where first a collinear structure forms, which then becomes non-collinear below T_{N2} . The previous work of the authors explained the AHE based on toroidal moments in triangular structure - so what is "in contrast" to triangular structure in these observations? What is the point then authors wanted to make by creating this fake "contrast"?

Based on the difference in magnetic structures reported in the present and previous studies, I also have a concern about Co stoichiometry or inter-site disorder in this system. The origin of the disparity needs to be understood and explained in order for the present "overriding" results to be treated seriously. Finally, the difference between the previous "toroidal" interpretation and the present "multi-Q" interpretation of the observed AHE must be clearly explained.

>130 "ordering is, therefore, the only possible scenario that can resolve this contradiction ..."
As noted above, specialize "only possible scenario" to "only possible scenario known to the present authors", or "we interpret this observation as ...". Again, the difference is not unlike that between "the best wine" and "the best wine I tried".

>144 "A uniform M_i^{tri} is obtained only when the Fourier components \vec{Q} of the three wave vectors \vec{Q} are orthogonal to each other"

>148-149 "nonuniform M, as illustrated in Supplementary Fig. 7. This is not realistic, ..."
Why is it "not realistic"? - such an assessment without any substantiation is untenable. Authors must provide some physical reason why they limit consideration to an equal-moment structures. In general, I see no reason for this at all. In the case of strongly under-saturated ordered moments, as in the material authors study, the ordered moments on different sublattices can have different lengths, or some sublattices can even be disordered, as in Supplementary fig. 7, d, or f. This would be specifically unsurprising in a Kondo lattice type system, which authors argue their material is, where screening of local moments by conduction electrons can easily be inhomogeneous. Authors rule this out as "non-realistic", but there seem to be little to no substantiation for such an assessment. In fact, partially disordered structures have been often postulated in the intermediate temperature range $T_{N2} < T < T_{N1}$ in triangular-lattice antiferromagnets.

>151 "This suggests that a transition from single-Q to triple-Q ordering occurs at T_{N2} ."
This assessment is unsubstantiated, - see comment above. Should be either removed, or substantiated by explaining how it follows from experimental data and why other, more traditional explanations such as transition from a partially- to a fully-ordered structure are ruled out.

>179 "reductions of the ordered moment ... is generally attributed to a partial delocalization of the magnetic moments. More specifically, in $\text{Co}_{1/3}\text{TaS}_2$, 180 interactions between Co local moments and itinerant electrons from Ta 5d band can lead to a partial screening of the local moments."
So, is it delocalization of Co 3d electrons, or Kondo-like screening of these by a 5d electrons of Ta?
- These look like two different scenarios, which authors somehow merge.

>188-189 "This naturally results in a magnetic order with ordering wave vectors corresponding to the three M points"
From Fig. 3c and 1f it follows that there are 6 equivalent M-points, not 3. The relation of the

stipulated 3-Q structure with the 6 M-points should be explained.

>211 "model successfully captures the tetrahedral ground state and the two-step transition process at"

On a mean-field level (looks like SI deals with classical spins?) – need to be specified if so.

> 207-209 "The accidental degeneracy is broken by effective four-spin exchange interactions that make the quadratic magnon mode gapped. The simplest example of a four-spin interaction favouring the triple-Q ordering is the bi-quadratic term $\lambda(S \cdot S)$ with $\lambda > 0$ "

I understand that discussion here is that of the J-t Kondo type Hamiltonian of Eq. (1) and its accidental degeneracy. The statement above then reads that biquadratic interaction should be added to Eq. (1). This does not look right. First, the effective spin Hamiltonian of supplementary Eq. (3) needs to be obtained from Eq. (1), and then it can be amended by higher-order terms. Finally, the effective spin Hamiltonian of Supplementary Eq. 3 can only have some accidental quadratic mode for a very specific set of parameters $J1/J2$ for which linear term zeros out.

>229-230 "a quadratic mode appears with weaker intensity as a line-shaped hexagon connecting six M points (see Supplementary Fig. 9)."

This is a misrepresentation/overinterpretation of the data. Firstly, it is not clear to me that feature authors discuss is indeed present in their data. Secondly, even if present, the only thing authors can say is that they observe some intensity which they ascribe to a stipulated quadratic mode under the provision that the mode is somehow phenomenologically gapped. Even such an interpretation would be a stretch. But stating that a quadratic mode is "observed" based on the data presented in Fig. 3 and Supplementary Fig. 9 is data misrepresentation at a borderline of scientific integrity.

>251-252 "Therefore, the sign change at $\pm k_Q$ represents the transition between tetrahedral orderings with positive and negative σ values"

Again, this is an interpretation and should be explicitly described as such by saying something like, "in our model, ..." or, "we interpret this as,..."

>254-255 "real-space Berry curvature of the tetrahedral triple-Q ordering should generate both the orbital ferromagnetic moment and spontaneous Hall conductivity"

Which orbital ferromagnetic moment? That of conduction electrons? If so, should be explicitly stated. Also, it should be clarified everywhere that the Berry phase of local-spin magnetic structure in the authors' scenario acts on conduction electrons via Kondo-type exchange coupling as in Eq. (1).

>263-265 "such a model cannot capture the sudden decrease of $12 \mu_B$ due to a meta-magnetic transition at $\pm k_Q$, indicating that this transition entirely changes the spin configuration." The decrease is very small, why would this suggest "entirely changed" spin configuration?

276-278 "In summary, we have discovered a tetrahedral triple-Q ordering in $\text{Co}_{1/3}\text{TaS}_2$. Our study provides a complete picture of how this exotic phase can be stabilized in the triangular metallic magnet $\text{Co}_{1/3}\text{TaS}_2$ and opens avenues for exploring chiral magnetic orderings with the potential for spontaneous integer quantum Hall effect"

This is the ultimate misrepresentation of the results. Tagging interpretation of the reported experimental observations a "discovery" is inappropriate and at a borderline of scientific integrity. Authors might have discovered unusual behaviors of the measured Hall conductivity, magnetic neutron scattering, or whatever other physical quantities they have measured and presented in the manuscript. They interpret these observation in a particular way. In their previous manuscript, same Hall results were interpreted differently. There is no reason to believe that other, alternate interpretations of these same results do not exist. At least this is not ruled out by the data presented in the manuscript.

> Supplementary p. 2. To guarantee an equal magnitude of \mathbf{M}_i for any i , the three Fourier component vectors $\Delta_{\mathbf{v}}$ should be perpendicular to each other ($\Delta_{\mathbf{v}} \perp \Delta_{\mathbf{v}'}$ for $\mathbf{v} \neq \mathbf{v}'$).

As I mentioned in comments to the main text, there is no substantiation for the above stated need to "guarantee an equal magnitude", especially for the Kondo-lattice-type model (but for local spins with strong fluctuations, too).

>Supplementary p. 4. the stripe order appears at high temperatures with enough thermal fluctuations to overcome the energy cost of positive K_{bq} . This result is precisely what we observed in $\text{Co}_{1/3}\text{TaS}_2$ and thus demonstrates the validity of our model.

Here, authors argue that theory is correct because they "observe" the single-Q to multi-Q transition in experiment. On the other hand, in the main text, the argument is exactly the opposite: the transition is claimed to be "observed" because it is what theory predicts. Looks like a simple fallacy to me?

> Supplementary p. 4. higher-order terms in the $1/S$ expansion is not included in our model, so K_{bq} in our results might be underestimated compared to their actual value.

But it is obtained from comparison with experiment and therefore accounts for everything?

> Supplementary p. 5. Symmetry-allowed single-ion anisotropy terms and the gapped linear magnon mode.

The origin of the single-ion anisotropy terms is atomic spin-orbit interaction, a relativistic effect, – and this should be included in the discussion. Consequently, different anisotropy terms authors discuss have different smallness in $(v/c)^2$ (fine structure constant) and therefore are of different magnitude. Authors should include the consequent consideration of the relative (un)importance of the anisotropy terms in their discussion here.

> Supplementary Fig. 3

The difference between ZFC and FC magnetization below $T_{\text{N}2}$ can indicate the formation of a tiny weak ferromagnetic moment, or formation of a single magnetic domain in a multi-domain structure. What is the mechanism here and how it fits in the multi-Q story authors present?

>Supplementary Table 1.

Co occupancy obtained in the Rietveld refinement of the powder XRD pattern indicates non-negligible off-stoichiometry. The material itself is very sensitive to the stoichiometry, though. How significant this effect is?

First Report of Referee #1

General Comment: I have read carefully the revised draft of the paper and replies to my previous questions. Overall, I think that the paper has improved considerably and the authors have addressed most of my concerns. In principle, I recommend this manuscript for publication after the authors have answered the following questions:

Reply: We appreciate the referee's careful consideration and positive evaluations of our manuscript. We have addressed the questions raised by the referee below.

Comment #1: The explanation of the ARPES data on the origin of the M-point triple-Q structure is problematic. In the Fermi surface nesting picture, the nesting susceptibility is proportional to the density of states at a certain nesting wavevector. In the case of hexagonal Fermi surfaces connecting the M-points, the nesting susceptibility should reach its maximum at the wavevectors connecting the edges of the FS hexagons, i.e., $(1/4 \ 1/4)$ and equivalent positions instead of $(1/2 \ 0)$ connecting the vertices, and the nesting susceptibility at the M point $(1/2 \ 0)$ should be at a saddle point instead of maximum. The authors should explain if the magnetic order is indeed driven by FS nesting, why is the magnetic order at $(1/2 \ 0)$ instead of $(1/4 \ 1/4)$.

Reply: Thank you for asking this critical question. It may appear problematic, but the explanation of the ARPES data is natural. In the first place, a perfectly nested Fermi surface (regular hexagon with corners at the M-points) leads to a \log^2 divergence of the magnetic susceptibility at the M-points. One of the logs arises from the logarithmically divergent density of states (Van Hove singularities) at the M points (note that M-points are connected by M wave vectors). The second logarithm arises from the perfect nesting (the edges of the hexagonal Fermi surfaces in different Brillouin zones are connected by M wave vectors). The point we make in the manuscript is that the measured Fermi surface seems to be well approximated by a regular hexagon whose corners are the M-points of the Brillouin zone (see Figure 3c). In summary, the nesting wave vector is not $(1/4, 1/4)$, but any M wave vector (see Refs. 12 and 30 of the new main text for more details).

Comment #2: What is the difference between the triple-Q and the stripe structure in terms of magnon dispersion? In principle, there are two differences. First, the stripe phase has two atoms per magnetic unit cell while the triple-Q phase has four, therefore the number of spin wave branches should be different, and the magnon dispersion at around Gamma and M should be identical for the triple-Q state. Second, the non-collinear spin structures will have more magnon-magnon coupling compared with the collinear one. Are the authors able to observe any of these effects in their INS spectra?

Reply: Thank you for the comment. In regard to the number of spin wave branches, unfortunately, technical limitations in the quality and data coverage of our INS dataset prevented us from giving a definitive answer. The main reason is that the superposition of the magnon dispersions for the three single-Q domains makes our analysis rather tricky: the detector setting we used only allows us access to the low-energy part of the magnon spectrum. Our INS data exhibits a rather considerable linewidth broadening consistent with the referee's second point. However, we cannot use this observation singly as evidence for the triple-Q structure as it could also be due to the metallic character of $\text{Co}_{1/3}\text{TaS}_2$ (magnon decay into particle-hole continuum).

Besides those points raised by the referee, another qualitative difference exists between the magnon dispersion of the triple-Q and stripe states. The triple-Q state exhibits isotropic velocity of the long-wavelength “quadratic” magnon mode (zero modes associated with the continuous degeneracy between single-Q, double-Q and triple-Q orderings that is obtained for isotropic Heisenberg interactions) in the k_x - k_y plane (circular ring patterns in Fig. R1). In contrast, the velocity of this low-energy mode is anisotropic for the stripe phase. (see Fig. R1 below).

Fig. R1 Long-wavelength magnon modes of the tetrahedral triple-Q and stripe states. The simulations in this figure only contain minimal resolution convolution and thus look different from the same calculation results shown in Fig. 3.

Comment #3: Have the authors compared the J1-J2-bq exchange model with J1-J2-J3 model? I understand the authors’ argument that the J3 should be small even for the RKKY interactions, but it would be interesting to see the effect of J3 on the magnetic structure because it connects the sublattices with the same spin direction for the triple-Q structure, and a ferromagnetic J3 will favor the triple-Q structure over the stripe one.

Reply: We thank the referee for the comment. With all due respect, however, J_3 does not favour the triple-Q structure over the stripe structure. This is because both magnetic structures possess a ferromagnetic configuration between the third nearest neighbours. More generally, isotropic bilinear exchange interactions cannot break the degeneracy between the stripe and tetrahedral triple-Q structure unless quantum fluctuations are included. Therefore, the four-spin interaction

terms (the bi-quadratic term is one example) must be added to stabilize the tetrahedral triple-Q ground state in the classical limit.

As we clarify in the new version of the SI, the $J_1+J_2+K_{bq}$ model is a minimal phenomenological model that we use to describe the low-energy part of a spectrum (This model can be used to describe the long-wavelength limit (a low energy excitation spectrum) of the problem under consideration because the corresponding non-linear sigma model has enough free parameters to control the shape and velocity of the Goldstone modes, as well as the gap of the accidental quadratic gapless mode.). As the reviewer noticed, given the metallic character of the material, a more realistic model should include longer-range exchange interactions and other types of 4-spin interactions (this is explicitly mentioned in the new version of the manuscript).

First Report of Referee #2

General Comment: Even though the revised manuscript is markedly different from the original submission, it suffers from many of the same problems that I noted in my first review. The most important one is the misrepresentation of the authors' interpretations of the experimental results presented in the manuscript as actual experimental "discoveries". In my first review I noted that no conclusive evidence was presented in the manuscript that would support the main claim that a triple-Q ordering was actually observed. In the revised manuscript, authors present more data that agrees with such an interpretation, but this still does not amount to "discovery" of "Tetrahedral triple-Q ordering ...", which is the title claim of the manuscript. I noted that tendency to this type of grandiose thinking has already created problem for the authors' scientific credibility, where they have now "unobserved" the title claim of their previous work, "Field-tunable toroidal moment and anomalous Hall effect in non-collinear antiferromagnetic Weyl semimetal $\text{Co}_{1/3}\text{TaS}_2$ ". Nevertheless, in the present version of the manuscript authors keep confusing their interpretations, surmisions, and assumptions with actual observations and experimental discoveries, reporting them as such.

Having said all of the above, I nevertheless am of an opinion that manuscript presents large amount of interesting high-quality data combining ARPES, neutron scattering, and bulk transport and magnetization measurement in a synergistic and very complementary fashion, and these results are of sufficiently high quality and impact to be publishable in Nature Communications. The results also lead to appealing interpretations, which, however (unfortunately), are not presented in a scientifically consistent and ethical manner. Hence, in order to be reconsidered for publication the manuscript needs to be significantly revised, including its title, abstract, and summary paragraph. Specifically, in the revised manuscript authors need to avoid grandiose claims where their hypotheses, surmisions, opinions, and interpretations are presented as "observations" and even "discoveries". Presently, such passages are

encountered in every other paragraph of the manuscript. Below, I append a long list of specific comments on the issues which caught my attention.

Reply: We thank the referee's enthusiasm for our work. Following the referee's suggestion, we clarified several statements throughout the manuscript. We have also addressed the comments raised by the referee one by one below.

Regrettably, however, we cannot accept the reviewer's unjustified criticism that the results are not presented in an "ethical manner". We judge this unsubstantiated claim is a serious accusation coming from an anonymous reviewer. While we accept that the manuscript should be better written for non-experts in the field, and we thank the reviewer for highlighting the parts that are not clear enough for the average reader, attributing these omissions to the lack of ethical values or scientific integrity is going too far. In our detailed replies below, we explain the scientific reasons why we disagree with some of the reviewer's comments.

Comment #1: Firstly, title needs to be changed to something like "large AHE in metallic antiferromagnet ...", or anything else, which however would faithfully describe experimental findings without misrepresenting an interpretation for an observation.

Reply: We changed the title: "Tetrahedral triple-**Q** magnetic ordering and large spontaneous Hall conductivity in metallic triangular antiferromagnet $\text{Co}_{1/3}\text{TaS}_2$." We believe that the nature of non-coplanar triple-**Q** in $\text{Co}_{1/3}\text{TaS}_2$ has been presented enough in our new manuscript.

Comment #2: >30(Abstract) "The triangular lattice antiferromagnet (TLAF) has been the standard paradigm ..."

>49 "The triangular lattice Heisenberg model is a textbook example that can host a diverse ..."

>220 "However, the finite value of J_c does not change the qualitative features of the 2D stripe and tetrahedral orderings."

The finite (and large) J_c dispossesses the system of many peculiarities that a 2D triangular lattice antiferromagnet has, and this needs to be clearly explained. Stacked triangular-lattice systems with antiferromagnetic inter-layer coupling are, in many respects, fundamentally different from peculiar 2D triangular-lattice antiferromagnet that "has been the standard paradigm" and a "textbook example" (3D-dimensionality strongly reduces quantum fluctuations, wave vector $\mathbf{Q} = (1/3, 1/3, 1/2)$ is different from $2\mathbf{Q} = (1/3, 1/3, 0)$, etc)

Reply: In general, the inter-layer coupling (J_c) can, in principle, change some aspects of the 2D triangular lattice antiferromagnet. However, the 3D structure of $\text{Co}_{1/3}\text{TaS}_2$ with antiferromagnetic J_c does not change the nature of the competing single-**Q** stripe and tetrahedral triple-**Q** orderings that are obtained in the 2D limit (the magnetic ordering of the

J_1, J_2 model persists for finite J_c). This is because of the inter-layer coupling geometry of $\text{Co}_{1/3}\text{TaS}_2$ governed by the 6_3 screw symmetry.

For example, divide each triangular layer into the 4-sublattices 1, 2, 3 and 4 of the tetrahedral triple- \mathbf{Q} phase. A given spin on sublattice 1 of layer N interacts with the remaining sublattices 2, 3 and 4 on layer $N+1$ via J_c . Similarly, sublattice 2 of layer N interacts with the other three sublattices of layer $N+1$. This is precisely the same geometry as the intralayer NN interaction (J_1). This argument also holds for the two-sublattice stripe phase: a given spin interacts with the two spins on the same sublattice and four spins on the other sublattices via J_1 and J_c . Consequently, for antiferromagnetic J_c , the magnetic ordering in layer $N+1$ is simply a repetition of the magnetic ordering in layer N with some translation (*i.e.*, the spin alignment between adjacent layers is NOT antiferromagnetic). The key features of these two orderings in 2D, such as accidental degeneracy, resulting gapless quadratic magnon modes, and non-zero net scalar spin chirality, remain unchanged.

Finally, statements like “*Stacked triangular-lattice systems with antiferromagnetic inter-layer coupling are, in many respects, fundamentally different from peculiar 2D triangular-lattice antiferromagnet*” are too generic. Of course, any real material is three-dimensional, and the 3D aspect may sometimes be relevant. However, that is not the case for the system under consideration (quantum fluctuations do not play an important role, and it does not develop an alternating spin configuration along the c -axis so that the competing orderings in the 2D limit have not changed).

Comment #3: 40 "Discovering magnetic orderings with novel properties and functionalities is one of the main goals of condensed matter physics"

Magnetic orderings, even with novel properties and functionalities, is a narrow subfield of condensed matter physics, so the above claim is an example of grandiose thinking of which there are plenty in the manuscript.

Reply: We respectfully disagree with this statement. Finding materials with novel functionalities has always been the primary motivation of condensed matter physics. Let’s review the topics that captured most of the attention of the condensed matter community over the last 40 years: quantum hall systems, high- T_c superconductors, manganites, multiferroics, skyrmion systems, heterostructures, and topological materials. What all these materials have in common is the potential for developing new technologies based on their functionalities. “Interesting but useless” is how Louis Néel famously described antiferromagnets materials for whose discovery he was awarded the 1970 Nobel. This simple anecdote illustrates the importance that physicists assign to discovering valuable materials with novel properties and functionalities.

We frankly do not see how that sentence is an example of “grandiose thinking”. An introduction aims to provide an adequate context, which can help the general reader to place the contribution in a bigger picture. This sentence, as well as the previous one about triangular magnets, is

simply an attempt to provide some context for general readers who are unfamiliar with the magnetic orderings observed in triangular magnets. The magnetically ordered state that we reported in the manuscript has been predicted for triangular antiferromagnets, and, as far as we know, it has never been reported before in any bulk material.

Comment #4: 46-47 "can produce a huge synthetic magnetic field that couples only to the orbital degrees of freedom of conduction electrons."

The mechanism (of Coulomb origin) should be clearly explained. Namely, that in the presence of exchange interaction with conduction electrons these magnetic structures can induce such and such effects on conduction electron motion.

Reply: For readers unfamiliar with it, we added another paragraph explaining this mechanism in the introduction and provided several references in the manuscript [see Refs. 3, 4, 12 and 13 of the new main text] where the mechanism is described in detail. We have also added references 3 and 4 at the end of the sentence listed by the referee.

With all due respect, however, we want to remind the referee that it is already a well-known and widely documented mechanism from more than 20 years ago. Please refer to the following papers, in addition to Refs 3, 4, 12 and 13:

- Taguchi, Y., Oohara, Y., Yoshizawa, H., Nagaosa, N., & Tokura, Y. *Science*, 291(5513), 2573-2576 (2001). → **Cited 921 times**
- Ye, J., Kim, Y. B., Millis, A. J., Shraiman, B. I., Majumdar, P., & Tešanović, Z. *Physical Review Letters*, 83(18), 3737 (1999). → **Cited 601 times**
- Nagaosa, N., Sinova, J., Onoda, S., MacDonald, A. H., & Ong, N. P.. *Reviews of Modern Physics*, 82(2), 1539 (2010) → **Cited 4204 times**

Comment #5: >50-51 "For any spin value, the generic ground state for nearest-neighbour (NN) antiferromagnetic interactions is the three-sublattice 120° structure shown in Fig. 1a "

"For any spin value" is redundant and of questionable validity. It should also be specified to which interactions this refers. In general case, spin-spirals are possible in row models and perhaps in models with general long-range interactions on triangular lattice.

Reply: The interactions are already specified in the sentence as "*nearest-neighbour (NN) antiferromagnetic interactions*". It is well known that the ground state of the *nearest-neighbour* Heisenberg model is a 120° structure for any value of the spin, including $S=1/2$. Therefore, we do not understand why the reviewer claims this statement is "redundant and of questionable

validity”. What is their justification for making such a claim? Can the reviewer provide references that contradict this statement?

Comment #6: 63-64 The spins of each sublattice point along the all-in or all-out principal directions of a regular tetrahedron. Co appears to be in an octahedral environment, and in the case of an ideal octahedron has 3 symmetry-equivalent axes that could define moment (spin) direction. The fourth "tetrahedral" axis is not consistent with crystallographic environment and does not look very natural. Partially disordered phases have been considered and also observed experimentally on triangular lattices. It appears from what follows that the observed magnetic Bragg peaks can be described by a partially-disordered structure where the "fourth" spin pointing along a "crystallographically unnatural" direction is simply not ordered. There is no explanation why this possibility is discarded (see also related comments below).

Reply: We added a comprehensive explanation in Supplementary Notes to clarify what kind of spin configurations are compatible with our experimental data. In response to the comment, we have brought some of them below and described them in more detail:

1) A possibility of a partially-disordered structure where “one of the four spins is disordered”.

We should point out that this case is utterly inconsistent with our experimental data. First, the referee should understand that the *antiferromagnetic* triple-**Q** spin configuration becomes coplanar if one of the four spins is disordered. There is NO way to satisfy the zero net magnetization condition ($\mathbf{S}_1 + \mathbf{S}_2 + \mathbf{S}_3 + \mathbf{S}_4 = 0$) if one of the four spins is disordered ($\langle \mathbf{S}_1 \rangle = 0$) and the others are non-coplanar ($\langle \mathbf{S}_2 \cdot (\mathbf{S}_3 \times \mathbf{S}_4) \rangle \neq 0$). This example is already illustrated in Supplementary Fig. 7f. However, as explained in the SI, our magnetic neutron diffraction data is only compatible with “non-coplanar” triple-**Q** ordering for $T < T_{N2}$. Therefore, *there is no partially-disordered triple-Q ordering that is compatible with the observed set of magnetic Bragg peaks.*

Furthermore, the partially disordered phase proposed by the referee cannot explain the observed anomalous Hall effect because it preserves the following symmetries that forbid finite $\sigma_{xy}(\mathbf{H} = 0)$:

- a) 6_3 screw symmetry combined with time reversal ($\equiv T \times (\tau_{0.5c} \times C_{2z})$),
- b) Two-fold rotation symmetry with the y-axis ($\equiv C_{2y}$).

Finally, it would be more helpful if the reviewer had provided specific references for their claims, “Partially disordered phases have been considered and observed experimentally on triangular lattices”. Without a reference, we cannot assess if the reported partially disordered phases have anything to do with the case under consideration. The statement “The fourth "tetrahedral" axis is inconsistent with the crystallographic environment and does not look very

natural” is just an opinion, which is not substantiated by facts. The *fact* is that, as we acknowledge in our manuscript, the tetrahedral ordering has been experimentally reported in hcp Mn monolayers on Re(0001), for which the fourth "tetrahedral" axis is also “not consistent” with the crystallographic environment (see Refs. 15 and 16 of the new manuscript). It is also a fact that the tetrahedral ordering is the most stable one of an extended J_1 - J_2 Heisenberg model (or any other Heisenberg model that exhibits an M-ordering ground state) by four-spin interactions if they are stronger than the anisotropic spin terms generated by the crystallographic environment in association with the relativistic spin-orbit coupling.

2) A possibility of a partially-distorted case referred to as “non-equilateral” tetrahedral ordering

Instead of the partially-disordered structure discussed above, a “partially distorted” tetrahedral spin configuration is also compatible with our transport and neutron diffraction data. In fact, this was discussed shortly in the previous version of SI. To further clarify this point, we have provided more elaborate explanations of this structure in the new SI, referred to as *non-equilateral* tetrahedral triple- \mathbf{Q} orderings. In response to the comment, we provide more explanations below.

We can express the tetrahedral ordering as a linear combination of three harmonic terms from $\mathbf{q}_m^1 = (1/2, 0, 0)$, $\mathbf{q}_m^2 = (-1/2, 1/2, 0)$, and $\mathbf{q}_m^3 = (0, -1/2, 0)$:

$$\mathbf{M}_i^{\text{tetr}} = \sum_{\nu=1}^3 \Delta_{\nu} \cos(\mathbf{q}_m^{\nu} \cdot \mathbf{r}_i).$$

The conditions: $\Delta_{\nu} \perp \Delta_{\nu'}$ for $\nu \neq \nu'$ and $|\Delta_{\nu}| = |\Delta_{\nu'}|$, lead to a high-symmetry four-sublattice structure, whose magnetic moments point along principal directions of a regular tetrahedron (= *equilateral*). However, the magnitude of three Fourier components does not have to be the same in general ($|\Delta_{\nu}| \neq |\Delta_{\nu'}|$). If not equal, the tetrahedron characterizing the four principal directions is no longer equilateral. Fig. R2 below shows one such example: $|\Delta_1| = |\Delta_2| = |\Delta_3|$ and $\sqrt{2} |\Delta_1| = |\Delta_2| = |\Delta_3|$. While both structures yield the same “powder-averaged” diffraction pattern and large $\sigma_{xy}(\mathbf{H} = 0)$, they are, in principle, distinguishable by a single-crystal diffraction measurement. This is because $|\Delta_{\nu}|^2$ is in proportion to the Bragg peak intensities on the M points corresponding to \mathbf{q}_m^{ν} . The bottom of Fig. R2 shows an excellent illustration of such.

Our single-crystal experimental results (Fig. 2c and 4e) indicate that the intensities of the three Bragg peaks from different \mathbf{q}_m^{ν} are identical within measurement errors, even under a magnetic field. Also, our simple phenomenological spin model (see the next note below), which reproduces the qualitative features of the measured INS data, predicts the equilateral tetrahedral ground state. These experimental and theoretical outcomes lead us to consider an

equilateral tetrahedral ordering as the primary candidate. Nevertheless, the non-equilateral tetrahedral ordering cannot be entirely excluded since the intensity of the three peaks can be accidentally identical due to the formation of magnetic domains. For example, three equally-populated magnetic domains of non-equilateral tetrahedral orderings, each constructed from $c|\Delta_1| = |\Delta_2| = |\Delta_3|$, $|\Delta_1| = c|\Delta_2| = |\Delta_3|$, and $|\Delta_1| = |\Delta_2| = c|\Delta_3|$ (c is an arbitrary coefficient within $0 < c < 1$), can accidentally give the same intensity. Finally, we note that the core properties of the tetrahedral ordering, such as scalar spin chirality and Berry's phase, are still present in both structures.

The possibility of a non-equilateral structure is now shortly mentioned in our new main text.

Fig. R2 Illustration of *equilateral* and *non-equilateral* tetrahedral orderings. The color and size of the red circles at the bottom depicts an intensity of the magnetic Bragg peak on a particular M point from each magnetic ordering.

Comment #7: 87-88 "This work reports the four-sublattice tetrahedral triple-Q ordering as the only viable scenario for the metallic triangular antiferromagnet Co_{1/3}TaS₂." Specialize "only viable scenario" to "only viable scenario known to the present authors". The difference is same as that between "the best food" and "the best food we tasted". The four-sublattice structure with one sublattice disordered still triple-Q. Is it at all important how many Q's are needed to describe it? Are there other partially ordered magnetic structures that would be consistent with the observed set of magnetic Bragg peaks?

Reply: As per the referee's suggestion, we revised the sentence accordingly. The answer to "partially ordered structures" can be found in the response to Comment #6.

Comment #8: >82-83 "where non-trivial band topology is induced in the absence of relativistic spin-orbit coupling (the noncoplanar configuration generates the large spin-orbit coupling)."

The spin-orbit interaction nomenclature refers to a relativistic effect (see eg H. A. Bethe, R. Jackiw, Intermediate Quantum Mechanics, Addison-Wesley (1997), pp. 152-178). The effects of exchange coupling between the ordered moments and conduction electrons are Coulomb effects, not spin-orbit interaction effects, and should be called what they are. This issue of unorthodox notations that are difficult to understand was a major unclarity in the previous manuscript, which I commented upon, and it still persists here. Authors might say that exchange effects can "transfer" spin chirality of magnetic structure to the orbital motion of conduction electrons leading to Hall effect etc, but not that they generate "spin-orbit coupling", which is conventionally understood as an atomic effect (see reference above).

Reply: While sympathetic to the referee's view, we fear that using the suggested convention might unnecessarily confuse readers. Indeed, it was after reading the first report by the same reviewer that we decided to add the word "relativistic" to distinguish the usual spin-orbit coupling (of relativistic origin) from the *effective* spin-orbit coupling that is generated by the Kondo exchange between conduction electrons and a non-coplanar texture of localized magnetic moments (see Ref. 4 of the new main text for more information). Note that while the Kondo exchange is an interaction that only involves the spin degree of freedom of the conduction electron, it produces an effective compact U(1) gauge field that couples to the orbital degrees of freedom of the same conduction electron when the underlying local moments form a non-coplanar structure. This is what we literally mean by "*effective spin-orbit* coupling".

We like to remind the referee that the spin and orbital degrees of freedom of an electron can be coupled even in the *absence of the relativistic effect*. This can be easily understood by considering the adiabatic limit of dominant Hund's of Kondo exchange (see <https://doi.org/10.1103/PhysRevB.101.024420> for more information) the spin of the electron is forced to remain parallel to the spin of the local moment. This constraint means that when the electron moves in a loop, it picks up a Berry phase equal to half of the solid angle spanned by the local moments enclosed by the loop. This phase cannot be distinguished from the Aharonov-Bohm phase that the same electron would pick up in the presence of a magnetic flux. Correspondingly, non-coplanar magnetic orderings lead to an *effective* magnetic field, whose flux in a given loop is proportional to the solid angle spanned by the local moments enclosed by the loop (a solid angle of 4π corresponds to one flux quantum). The remarkable aspect of the tetrahedral ordering reported in our manuscript is that the effective field in the adiabatic

limit leads to a flux quantum per 4 triangular plaquettes because the 3 spins of one triangle span $\frac{1}{4}$ of the full solid angle (4π) of the sphere.

It is important to understand this point to appreciate the significance of finding the tetrahedral magnetic ordering in a 3D material. Finally, the relativistic spin-orbit effect is not an “atomic effect” (for more information, see J. Frohlich and U. M. Studer, Rev. Mod. Phys. 65, 733 (1993)).

Comment #9: >100-101 "previous neutron diffraction study reported the ordering wave vector $\mathbf{q}_m = (1/3, 1/3, 0)$ characteristic of 120° ordering. In contrast, the latest experimental study of the single-crystal $\text{Co}_{1/3}\text{TaS}_2$ reported two antiferromagnetic phase transitions at $T_{N1} = 38$ K and $T_{N2} = 26.5$ K (Fig. 1h), as well as a significant anomalous Hall effect (AHE) comparable to that of ferromagnets below T_{N2} ."

What is "in contrast" here? Triangular ordering more often than not comes with two Neel temperatures, where first a collinear structure forms, which then becomes non-collinear below T_{N2} . The previous work of the authors explained the AHE based on toroidal moments in triangular structure - so what is "in contrast" to triangular structure in these observations? What is the point then authors wanted to make by creating this fake "contrast"?

Based on the difference in magnetic structures reported in the present and previous studies, I also have a concern about Co stoichiometry or inter-site disorder in this system. The origin of the disparity needs to be understood and explained in order for the present "overriding" results to be treated seriously. Finally, the difference between the previous "toroidal" interpretation and the present "multi-Q" interpretation of the observed AHE must be clearly explained.

Reply: Following the referee’s comment, we removed the words “in contrast” (it was simply a usage of a wrong conjunction) and revised the text:

“Previous studies on $\text{Co}_{1/3}\text{TaS}_2$ in the 1980s reported the bulk properties of a metallic antiferromagnet with $S=3/2$ (a high-spin d^7 configuration of Co^{2+}),²³⁻²⁵ including a neutron diffraction study that reported an ordering wave vector $\mathbf{q}_m = (1/3, 1/3, 0)$ characteristic of a 120° ordering²⁵. More recently, an experimental study on single-crystal $\text{Co}_{1/3}\text{TaS}_2$ observed a significant anomalous Hall effect (AHE) comparable to those in ferromagnets below 26.5 K, which is the second transition temperature (T_{N2}) of the two antiferromagnetic phase transitions at $T_{N1} = 38$ K and $T_{N2} = 26.5$ K (see Fig. 1h)²⁶. Based on the 120° ordering reported in Ref. ²⁵ and a symmetry argument, the authors of Ref. ²⁶ suggested that the observed AHE, $\sigma_{xy}(\mathbf{H} = 0) \neq 0$, can be attributed to a ferroic order of cluster toroidal dipole moments. However, our latest neutron scattering data reported in this work reveals an entirely different picture: $\text{Co}_{1/3}\text{TaS}_2$ has a magnetic structure with ordering wave vectors of the M-points ($\mathbf{q}_m = (1/2, 0, 0)$ and symmetry-related vectors) instead of $\mathbf{q}_m = (1/3, 1/3, 0)$. This new observation forced us

to conduct a more extensive investigation and come up with a scenario consistent with all the experimental facts..”

Let us clarify the main difference between the present paper and the earlier one (see also our response to comment #21). Previously, by accepting the $\mathbf{q}_m = (1/3, 1/3, 0)$ reported in the 1980s, we came to the “toroidal” interpretation to explain the observed AHE. Such interpretation was the only way to explain non-zero $\sigma_{xy}(\mathbf{H} = 0)$ under the magnetic ordering with $\mathbf{q}_m = (1/3, 1/3, 0)$ based on symmetry arguments. In this picture, the source of Berry curvature was the electron band structure – hourglass Weyl points protected by 6_3 screw symmetry.

However, while doing the subsequent works, we began questioning the old report of the $\mathbf{q}_m = (1/3, 1/3, 0)$ reported in the 1980s. And we decided to do our own neutron diffraction experiments: both powder and single crystal. For this new work, we have discovered that the correct magnetic structure has a different ordering wavevector $\mathbf{q}_m = (1/2, 0, 0)$. This new discovery forced us to reconsider the underlying physics from the beginning; the nature of $\mathbf{q}_m = (1/2, 0, 0)$ is entirely different from that of $\mathbf{q}_m = (1/3, 1/3, 0)$. Using this new magnetic ordering wavevector, we conclude in this paper that the Berry curvature of the non-coplanar “multi- \mathbf{Q} ” structure is the source of the observed AHE. Note that the symmetry-protected Weyl points are no longer valid in the case of $\mathbf{q}_m = (1/2, 0, 0)$ as this magnetic ordering breaks the 6_3 screw symmetry.

Finally, we would like to explain the origin of the inconsistency between the \mathbf{q}_m reported in the cited reference and in our work. Indeed, the Co stoichiometry is a crucial factor that changes the magnetic ordering wave vector in $\text{Co}_{1/3}\text{TaS}_2$. However, even after a thorough investigation of Co_xTaS_2 with $0.299 < x < 0.34$, we could not find any sample that exhibits $\mathbf{q}_m = (1/3, 1/3, 0)$ ordering; instead, $\mathbf{q}_m = (1/3, 0, 0)$ was only observed in over-doped $\text{Co}_{1/3}\text{TaS}_2$, in addition to $\mathbf{q}_m = (1/2, 0, 0)$ – see our response to comment #27. We should also point out that the $\mathbf{q}_m = (1/3, 1/3, 0)$ does not appear in the phase diagram (see Supplementary Fig. 8b) if the antiferromagnetic inter-layer coupling J_c is comparable to J_1 , which is the case of $\text{Co}_{1/3}\text{TaS}_2$ as confirmed by our inelastic neutron scattering measurement. On the other hand, a large region of $\mathbf{q}_m = (1/2, 0, 0)$ and $\mathbf{q}_m = (1/3, 0, 0)$ phases are found in the phase diagram of J_1 – J_2 – J_3 – J_c model. This leads us to carefully suggest the possibility that the previous report of $\mathbf{q}_m = (1/3, 1/3, 0)$ ordering is simply incorrect, presumably due to some errors in analyzing their neutron diffraction data.

Comment #10: >130 "ordering is, therefore, the only possible scenario that can resolve this contradiction ..."

As noted above, specialize "only possible scenario" to "only possible scenario known to the present authors", or "we interpret this observation as ...". Again, the difference is not unlike that between "the best wine" and "the best wine I tried".

Reply: Thank you. We revised the sentence accordingly.

Comment #11: >144 " A uniform $|\mathbf{M}_i^{\text{tri}}|$ is obtained only when the Fourier components Δ_ν of the three wave vectors \mathbf{q}_m^ν are orthogonal to each other¹⁰, *i.e.*, $\mathbf{M}_i^{\text{tri}} = \sum_{\nu=1,3} \Delta_\nu \cos(\mathbf{q}_m^\nu \cdot \mathbf{r}_i)$ with $\Delta_\nu \perp \Delta_{\nu'}$ for $\nu \neq \nu'$."
>148-149 "nonuniform M, as illustrated in Supplementary Fig. 7. This is not realistic, ..."

Why is it "not realistic"? – such an assessment without any substantiation is untenable. Authors must provide some physical reason why they limit consideration to an equal-moment structures. In general, I see no reason for this at all. In the case of strongly under-saturated ordered moments, as in the material authors study, the ordered moments on different sublattices can have different lengths, or some sublattices can even be disordered, as in Supplementary fig. 7, d, or f. This would be specifically unsurprising in a Kondo lattice type system, which authors argue their material is, where screening of local moments by conduction electrons can easily be inhomogeneous. Authors rule this out as "non-realistic", but there seem to be little to no substantiation for such an assessment. In fact, partially disordered structures have been often postulated in the intermediate temperature range $T_{N2} < T < T_{N1}$ in triangular-lattice antiferromagnets.

Reply: Thank you for the comment. We urge the referee to see the statement in the whole context. The ARPES data shown in the manuscript indicates that the 3- \mathbf{Q} ordering is a Fermi surface effect; the Kondo lattice system is in the weak-coupling regime and the effective RKKY interaction induces the magnetic ordering. Therefore, Kondo screening is expected to be relatively small in this regime, where any reduction of the moment length will come at the expense of an RKKY energy cost. Note that the 3- \mathbf{Q} *collinear* ordering – the 3- \mathbf{Q} structure compatible with the neutron diffraction profile in $T_{N2} < T < T_{N1}$ – leads to a considerable reduction of the ordered moment, such that the moment on three sublattices is 1/3 of the moment on the 4th sublattice (see Fig. 7 of the supplementary information). This moment reduction is too big to be explained by Kondo screening in a regime dominated by the RKKY interaction.

Also, the collinear triple- \mathbf{Q} ordering allows for finite $\sigma_{xy}(\mathbf{H} = 0)$ by symmetry, while the observed value is "precisely" zero within our measurement error in $T_{N2} < T < T_{N1}$. On the other hand, the single- \mathbf{Q} ordering shown in Fig. 2g explains the absence of $M_z(\mathbf{H} = 0)$ and $\sigma_{xy}(\mathbf{H} = 0)$ at $T_{N2} < T < T_{N1}$ because these quantities are forbidden by the $\tau_{1a}T$ symmetry. To avoid confusion, we have modified the relevant sentences in the main text.

Comment #12: >151 "This suggests that a transition from single- \mathbf{Q} to triple- \mathbf{Q} ordering occurs at T_{N2} ."

This assessment is unsubstantiated, - see comment above. Should be either removed, or substantiated by explaining how it follows from experimental data and why other, more traditional explanations such as transition from a partially- to a fully-ordered structure are ruled out.

Reply: Thank you for the comment. Please see our detailed response to comments #6 and #11.

Comment #13: >179 "reductions of the ordered moment ... is generally attributed to a partial delocalization of the magnetic moments. More specifically, in Co_{1/3}TaS₂, interactions between Co local moments and itinerant electrons from Ta 5d band can lead to a partial screening of the local moments."

So, is it delocalization of Co 3d electrons, or Kondo-like screening of these by a 5d electrons of Ta? – These look like two different scenarios, which authors somehow merge.

Reply: We are sorry for the confusion, and we did not intend to mix up the two scenarios. We meant that for *d*-electron ions, like Co, there are two potential sources of magnetic moment reduction. The first one corresponds to charge fluctuations (or partial delocalization), which are not small for d-electron systems. The second one is the usual Kondo screening. Based on the existing data, we cannot separate both contributions. To clarify the message, we changed the sentence to:

“~is generally attributed to a partial delocalization of the magnetic moments. In addition, the effective Kondo exchange between the Co moments and the conduction electrons can also produce a partial screening of the local moments.”

Comment #14: 188-189 "This naturally results in a magnetic order with ordering wave vectors corresponding to the three M points"

From Fig. 3c and 1f it follows that there are 6 equivalent M-points, not 3. The relation of the stipulated 3-Q structure with the 6 M-points should be explained.

Reply: The **M** points are invariant under spatial inversion $\mathbf{Q} \rightarrow -\mathbf{Q}$ because they are equal to half of a reciprocal lattice vector. For instance $(-1/2, 0, 0) = (-1, 0, 0) + (1/2, 0, 0)$. Therefore, half of the “6 equivalent M points” that the referee mentioned are the same as the other half without any meaningful difference.

Comment #15: 211 "model successfully captures the tetrahedral ground state and the two-step transition process at"

On a mean-field level (looks like SI deals with classical spins?) – need to be specified if so.

Reply: As we already explained in the SI, the results were obtained by classical Monte-Carlo simulations. Following the referee's suggestion, we have specified the method in the sentence.

Comment #16: > 207-209 "The accidental degeneracy is broken by effective four-spin exchange interactions that make the quadratic magnon mode gapped. The simplest example of a four-spin interaction favouring the triple-Q ordering is the bi-quadratic term $K_{\text{bq}}(\mathbf{S}_i \cdot \mathbf{S}_j)^2$ with $K_{\text{bq}} > 0$."

I understand that discussion here is that of the J-t Kondo type Hamiltonian of Eq. (1) and its accidental degeneracy. The statement above then reads that biquadratic interaction should be added to Eq. (1). This does not look right. First, the effective spin Hamiltonian of supplementary Eq. (3) needs to be obtained from Eq. (1), and then it can be amended by higher-order terms. Finally, the effective spin Hamiltonian of Supplementary Eq. 3 can only have some accidental quadratic mode for a very specific set of parameters J_1/J_2 for which linear term zeros out.

Reply: If we assume that a Kondo lattice model describes the material, the effective low-energy model will naturally include an RKKY contribution with longer range interactions (beyond second neighbours) and different types of 4-spin interactions (see Eq.(9) of [arXiv:2212.09796](https://arxiv.org/abs/2212.09796)). Since we do not have a microscopic model for this material (a realistic microscopic Kondo lattice model should include all the conduction bands and hybridizations), we opted for using a minimal *phenomenological* effective spin model that contains all the essential ingredients to describe the magnetic properties in the long wavelength limit. More specifically, a model whose long wavelength limit (non-linear sigma model) includes all the parameters necessary to describe the low-energy excitation spectrum. This is now more explicit in the new version of the supplementary material.

The quadratic zero mode is present for a pure Heisenberg model $H_{\text{Heis}}(J_1 J_2 J_3, \dots)$ with *any* set of exchange parameters $(J_1 J_2 J_3, \dots)$, such that the Fourier transform of the exchange interaction has global minima at the M points. The simple reason is that any of the classical magnetic orderings $\mathbf{S}_i = \sum_{\nu=1,3} \Delta_\nu \cos(\mathbf{q}_m^\nu \cdot \mathbf{r}_i)$ with $\Delta_\nu \perp \Delta_{\nu'}$ for $\nu \neq \nu'$ and $\sum_{\nu=1,3} |\Delta_\nu|^2 = S^2$ is a ground state of the classical spin Hamiltonian (note that this expression includes collinear, coplanar and non-coplanar orderings because some of the vector amplitudes Δ_ν can vanish). The simple phenomenological J_1 - J_2 -biquadratic model captures the zero quadratic and Goldstone modes of the single-Q and triple-Q orderings in the long wavelength limit.

Comment #17: 229-230 "a quadratic mode appears with weaker intensity as a line-shaped hexagon connecting six M points (see Supplementary Fig. 9)."

This is a misrepresentation/overinterpretation of the data. Firstly, it is not clear to me that feature authors discuss is indeed present in their data. Secondly, even if present, the only thing authors can say is that they observe some intensity which they ascribe

to a stipulated quadratic mode under the provision that the mode is somehow phenomenologically gapped. Even such an interpretation would be a stretch. But stating that a quadratic mode is “observed” based on the data presented in Fig. 3 and Supplementary Fig. 9 is data misrepresentation at a borderline of scientific integrity.

Reply: Fig. R3 below are the same plots from Fig. 3f ($E = 2$ & 2.5 meV) but with different colour bar scales. There are clear signals connecting the six M points of the Brillouin zone, in addition to the six bright circular signals centered at the M points that correspond to linear magnon modes.

While we have toned down our statement in the new version of the text, we would like to stress that when excluding the linear magnon modes (which we can clearly identify in the figure below or in Fig 3f as six circular features corresponding to cross-sections of linear dispersion cones), the remaining low-energy magnon modes can only be attributed to a zero energy mode (= quadratic mode) that is predicted by spin-wave theory for *any* isotropic Heisenberg model whose classical ground state exhibits M-ordering. If we apply the “rigorous” criterion adopted by the reviewer, we should not interpret the circles centered at the M points as “Goldstone modes”. That would also be a “misrepresentation/overinterpretation” of the data because to identify those low-energy modes with Goldstone modes, we need to accept that the system is well-described by a quasi-isotropic (SU(2) invariant) model. The reviewer does not question that interpretation because he/she seems familiar with this concept. If we accept that the relevant spin model is well described (to a very good approximation) by an isotropic Heisenberg model *with relatively small 4-spin interactions and anisotropy terms*, the presence of the low-energy quadratic zero modes *is guaranteed* because of the continuous degeneracy that is explained in the previous point.

Fig. R3 Constant-energy cuts of the INS data measured at 5 K ($< T_{N2}$).

Comment #18: >251-252 "Therefore, the sign change at $\pm H_{c1}$ represents the transition between tetrahedral orderings with positive and negative χ_{ijk} values "

Again, this is an interpretation and should be explicitly described as such by saying something like, “in our model, ...” or, “we interpret this as,...”.

Reply: We revised the sentence accordingly.

Comment #19: 254-255 "real-space Berry curvature of the tetrahedral triple-Q ordering should generate both the orbital ferromagnetic moment and spontaneous Hall conductivity"

Which orbital ferromagnetic moment? That of conduction electrons? If so, should be explicitly stated. Also, it should be clarified everywhere that the Berry phase of local-spin magnetic structure in the authors' scenario acts on conduction electrons via Kondo-type exchange coupling as in Eq. (1).

Reply: We have now clarified them in the revised manuscript. Also, please see our response to Comments #4, #8, and #26.

Comment #20: >263-265 " However, such a model cannot capture the sudden decrease of σ_{xy}^{AHE} due to a meta-magnetic transition at $\pm H_{c2}$, indicating that this transition entirely changes the spin configuration."

The decrease is very small, why would this suggest "entirely changed" spin configuration?

Reply: We deleted the word 'entirely' from our sentence.

Comment #21: 276-278 "In summary, we have discovered a tetrahedral triple-Q ordering in Co_{1/3}TaS₂. Our study provides a complete picture of how this exotic phase can be stabilized in the triangular metallic magnet Co_{1/3}TaS₂ and opens avenues for exploring chiral magnetic orderings with the potential for spontaneous integer quantum Hall effect"

This is the ultimate misrepresentation of the results. Tagging interpretation of the reported experimental observations a "discovery" is inappropriate and at a borderline of scientific integrity. Authors might have discovered unusual behaviors of the measured Hall conductivity, magnetic neutron scattering, or whatever other physical quantities they have measured and presented in the manuscript. They interpret these observation in a particular way. In their previous manuscript, same Hall results were interpreted differently. There is no reason to believe that other, alternate interpretations of these same results do not exist. At least this is not ruled out by the data presented in the manuscript.

Reply: While we have toned down our main text, we disagree with the referee's comment and want to clarify misunderstandings.

“Tagging interpretation of the reported experimental observations a “discovery” is inappropriate and at a borderline of scientific integrity. Authors might have discovered unusual behaviors of the measured Hall conductivity, magnetic neutron scattering, or whatever other physical quantities they have measured and presented in the manuscript. They interpret these observation in a particular way”

→ As described in the new version of the manuscript and in the reply, a combination of neutron diffraction measurements and transport anomalies exclude single- \mathbf{Q} , double- \mathbf{Q} , and collinear/co-planar triple- \mathbf{Q} orderings (a partially-disordered triple- \mathbf{Q} , where one of the four spins are disordered, also belongs to this case) for $T < T_{N2}$. The only remaining possibility is non-coplanar triple- \mathbf{Q} , which results in a four-sublattice tetrahedral spin configuration. Even though a tetrahedral configuration with a certain amount of distortion (= the non-equilateral case) is compatible with our experimental data (and we have NOT been ignoring this possibility), it does not compromise the core features of the equilateral tetrahedral ordering. Honestly, we do not see what other interpretations should be further considered. Moreover, we are suggesting a simple theoretical conjecture of how the equilateral tetrahedral spin configuration can become the ground state in $\text{Co}_{1/3}\text{TaS}_2$. The proposed model is consistent with the electronic structure and the predicted long-wavelength spin dynamics is consistent with the dispersion and the intensities measured by inelastic neutron scattering.

“In their previous manuscript, same Hall results were interpreted differently. There is no reason to believe that other, alternate interpretations of these same results do not exist.”

→ We would like to stress again that the change of our interpretation was *not* due to our misrepresentation of the result but simply because the previous neutron diffraction study we referred to turns out to be *incorrect*. As our response to comment #9 explained, our previous interpretation was made with the wrong magnetic wavevector reported in the 80’s – $\mathbf{q}_m = (1/3, 1/3, 0)$. Unless one denies this observation, the interpretation based on a ferro-toroidal configuration was the only way that can explain non-zero $\sigma_{xy}(\mathbf{H} = 0)$ under the magnetic ordering of $\mathbf{q}_m = (1/3, 1/3, 0)$ due to symmetry. On the other hand, this work has been prompted by our experimental observation of the new magnetic wavevector; $\mathbf{q}_m = (1/2, 0, 0)$. Indeed, the previous scenario based on $\mathbf{q}_m = (1/3, 1/3, 0)$ is no longer valid as the spin configuration is completely different from the case of $\mathbf{q}_m = (1/2, 0, 0)$. Thus, as the referee agreed, we have conducted extensive experimental and theoretical works to suggest a new mechanism compatible with $\mathbf{q}_m = (1/2, 0, 0)$ and confirm its validity.

So once again, the statement “while the authors chose a particular way of interpretation among many other possibilities in the previous manuscript, they have now changed their interpretation to another one” is not true at all. Instead, each manuscript has highlighted the magnetic ordering consistent with the large spontaneous Hall conductivity based on the two different \mathbf{q}_m , and the other possibilities are logically excluded by symmetry arguments. This is

how science works: new measurements sometimes correct previous results and lead to new interpretations and discoveries. The scientific progress over many centuries include numerous examples, where prominent scientists had to retract and change their interpretation of the experimental data. However, nobody has ever questioned their ‘*scientific integrity*’ for changing their minds.

Comment #22: > Supplementary p. 2. “To guarantee an equal magnitude of \mathbf{M}_i tri for any i , the three Fourier component vectors $\Delta\mathbf{v}$ should be perpendicular to each other ($\Delta\mathbf{v} \perp \Delta\mathbf{v}'$ for $v \neq v'$).”

As I mentioned in comments to the main text, there is no substantiation for the above stated need to “guarantee an equal magnitude”, especially for the Kondo-lattice-type model (but for local spins with strong fluctuations, too).

Reply: Please find the response to the comment #11.

Comment #23: >Supplementary p. 4. the stripe order appears at high temperatures with enough thermal fluctuations to overcome the energy cost of positive K_{bq} . This result is precisely what we observed in $\text{Co}_{1/3}\text{TaS}_2$ and thus demonstrates the validity of our model.

Here, authors argue that theory is correct because they “observe” the single-Q to multi-Q transition in experiment. On the other hand, in the main text, the argument is exactly the opposite: the transition is claimed to be “observed” because it is what theory predicts. Looks like a simple fallacy to me?

Reply: We have revised the relevant sentences to avoid misunderstanding. We are not claiming that “*the transition is claimed to be “observed” because it is what theory predicts.*”. As explained in the main text and response to comment # 11, our combined experimental data from neutron diffraction and transport measurements indicate that the single-Q ordering is more plausible than a *collinear* double-Q/triple-Q ordering in $T_{N2} < T < T_{N1}$. Also, the combined data set (neutron scattering + anomalous transport) indicates that the low-temperature ordered phase is a non-coplanar 3-Q magnetic ordering. These conclusions do not depend on any theoretical model.

As we explained above, the J_1 – J_2 – K_{bq} model is introduced as a phenomenological model to see if it provides a consistent description of the long wavelength physics of the material (low energy magnon spectrum). This simple model reproduces the observed low-energy features of the measured single-magnon dispersion and produces a thermodynamic phase diagram consistent with the data. Based on these observations, we infer that the model adequately describes the material.

Comment #24: > Supplementary p. 4. “higher-order terms in the $1/S$ expansion is not included in our model, so K_{bq} in our results might be underestimated compared to their actual value.” But it is obtained from comparison with experiment and therefore accounts for everything?

Reply: This higher-order $1/S$ correction is based on the very recent finding by one of the authors of this manuscript, which has not been included in previous works that deal with single-ion anisotropy or bi-quadratic interactions. In general, the amplitudes of terms that are non-linear in the spin operators on a given site, such as single-ion anisotropies or biquadratic interactions, must be renormalized relative to the values that are obtained from fitting the data with linear spin wave theory. For more information, see <https://arxiv.org/abs/2304.03874>.

The sentence above simply states that this renormalization was not included in our calculation. Thus, one should be cautious when comparing the single-ion anisotropy or bi-quadratic terms from different studies. For completeness, in the new version of the manuscript, we also included the value of K_{bq} that is obtained after applying the renormalization coefficient.

Comment #25: > Supplementary p. 5. “Symmetry-allowed single-ion anisotropy terms and the gapped linear magnon mode.”

The origin of the single-ion anisotropy terms is atomic spin-orbit interaction, a relativistic effect, – and this should be included in the discussion. Consequently, different anisotropy terms authors discuss have different smallness in $(v/c)^2$ (fine structure constant) and therefore are of different magnitude. Authors should include the consequent consideration of the relative (un)importance of the anisotropy terms in their discussion here.

Reply: We removed the discussion of single-ion anisotropy because quartic and sixth-order single-ion anisotropy terms are, in fact, inconsistent with the assumption of spin-3/2 for the Co^{2+} ions. While this is a conjecture based on the magnetic moment value extracted from the high-temperature susceptibility, it is better for clarity to keep the model consistent with this conjecture. The only allowed single-ion anisotropy term $K(S^z)^2$ for $S=3/2$ does not open a gap at the linear spin-wave level. Correspondingly, the observed gap must be attributed to higher order $1/S$ corrections in the presence of single-ion and exchange anisotropy terms that break all the continuous symmetries of the spin model [note that $(S^z)^2$ leaves a residual $U(1)$ symmetry spontaneously broken by the tetrahedral ordering].

Comment #26: > Supplementary Fig. 3

The difference between ZFC and FC magnetization below T_{N2} can indicate the formation of a tiny weak ferromagnetic moment, or formation of a single magnetic domain in a multi-domain structure. What is the mechanism here and how it fits in the multi-Q story authors present?

Reply: As the referee pointed out, there appears to be a weak ferromagnetic moment along the c -axis below T_{N2} . This spontaneous component can also be seen in Fig. 2f and Fig. 4c. The mechanism is straightforward: the tetrahedral ordering is ferro-chiral (same scalar chirality on each triangular plaquette). As such, the fictitious magnetic field produced by this non-coplanar ordering is uniform and only couples to the orbital degrees of freedom of the conduction electrons (orbital ferromagnet). The weak ferromagnetic moment is then attributed to the orbital magnetization (conduction electrons) induced by this uniform fictitious field.

Comment #27: >Supplementary Table 1.

Co occupancy obtained in the Rietveld refinement of the powder XRD pattern indicates non-negligible off-stoichiometry. The material itself is very sensitive to the stoichiometry, though. How significant this effect is?

Reply: We confirmed that the overall bulk properties (such as $\mathbf{q}_m = (1/2, 0, 0)$, two successive phase transitions, anomalous Hall effect, and weak ferromagnetic moment) are still present for Co_xTaS_2 within $0.299(4) < x < 0.325(4)$. However, considering the inevitable uncertainty of estimating a composition, we found that diagnosing the sample by measuring its transition temperature T_{N2} from the magnetometry is more accurate. Unless the off-stoichiometry is significant, T_{N2} has been measured as 26.5 K consistently so far. As far as the sample shows this value, no noticeable difference was observed in the bulk properties.

All experimental data in our manuscript, including the powder XRD, were collected from the sample that has $T_{N2} = 26.5$ K. Based on our empirical idea, this roughly corresponds to less than ~8 % Co vacancy (i.e. Co_xTaS_2 with $x > 0.306(4)$). For the sample with $x < 0.306$, T_{N2} becomes lower than 26.5 K, such as 22 K .

Reviewers' Comments:

Reviewer #1:

Remarks to the Author:

I read the revised draft and replies by the authors. I am happy with the replies and recommend the paper for publication.

Reviewer #2:

Remarks to the Author:

In the revised manuscript authors have addressed large fraction of my comments and suggestions. While the ensuing revisions have notably improved the manuscript, a number of deficiencies still remain, which leave me less than convinced in the validity of the results and interpretations that authors present. Hence, while I maintain that manuscript presents large number of interesting experimental results which are potentially of sufficient importance to warrant publication, I cannot recommend acceptance of the manuscript in its present form.

The most important problem of the manuscript is the discrepancy of the presented magnetic structure refinement with that previously reported in Ref. 25. It is this discrepancy that makes grounds for the new triple-Q scenario compared to the "toroidal" scenario previously "discovered" by some of the authors. It is unclear, what is the reason for the observed different magnetic structures. In their rebuttal letter, authors seem to suggest that authors of Ref 25 simply made a mistake in magnetic structure refinement. In the manuscript, however, authors sweep this issue under the rug and write,

Line 116-118 >This new observation forced us to conduct a more extensive investigation and come up with a scenario consistent with all the experimental facts.

It is obvious that the scenario proposed in the manuscript is not consistent with the experimental facts reported in Ref. 25. Rather than outright discarding the previous work, authors actually have the burden of proof to explain why their results are different and why reader should trust their results and not those of Ref. 25. In fact, judging by the scientific track record of the authors of both papers, this reviewer would put more trust in the refinement of Ref. 25 (Jane Brown's group).

There are also other explanations which question the scenario proposed in the manuscript.

Perhaps, the sample used in the present study and that of Ref 25 are somehow different? Perhaps, they differ in stoichiometry and/or degree of structural disorder? Here, I note that authors' own Xray structural refinement indicates lower occupancy of the Co sites (and this is consistent with neutron refinement) and a very large displacement parameter for Co, up to an order of magnitude larger than that for other atoms. Can these imperfections impact the authors' observations and explain their divergence from the previous results of Ref. 25?

Additionally, a comparison of the neutron powder diffraction patterns at 3K and 60K shown in Fig. 2a reveals that aside from magnetic reflections, also a couple of structural reflections disappear at 60K: peak at $\sim 1.7 \text{ \AA}^{-1}$, to the right of $(1/2 \ 0 \ 3)$ magnetic reflection, also peaks at ~ 2.3 and $\sim 2.7 \text{ \AA}^{-1}$. If these peaks are not magnetic (and they are not marked as magnetic, but are marked as structural), then the magnetic structure that authors have refined is in doubt. If these peaks are structural indeed, then the transition is magnetostructural in origin and the results of the corresponding structural change need to be understood. This is also corroborated by the first-order type singularity in magnetization (Fig. 1h) at the transition temperature which authors tag as TN2. It is not implausible that some lowering of the magnetostructural symmetry indicated by these peaks might explain the observed AHE in the weakly ferromagnetic phase at low T? Authors need to explain and disentangle these observations.

Finally, the intercalated dichalcogenide materials family is known to be extremely sensitive to the stoichiometry of the intercalating 3D metal. The difference of ~ 0.01 - 0.02 in the vicinity of $1/3$, ie between 0.31 and 0.33 might drastically change the system's behavior. That authors find this order of difference between their EDX/ICP chemical analysis results and neutron/Xray refinement of the corresponding site occupancies might indicate substantial degree of disorder in the Co position, which in turn might be crucial for the distinct magnetism in the sample authors studied compared to Ref 25. This is also consistent with large displacement parameter found in Xray

diffraction.

The related comment is,

On line 161 authors write: "only non-coplanar triple-Q orderings corresponding to equilateral ($|\vec{Q}_1| = |\vec{Q}_2| = |\vec{Q}_3|$, Fig. 2h) or non-equilateral ($|\vec{Q}_1| \neq |\vec{Q}_2| = |\vec{Q}_3|$) tetrahedral configurations are consistent with our Rietveld refinement of neutron diffraction data"

Why is the chi-squared of the neutron refinement (Suppl Table 4) so bad, an order-of-magnitude higher than that of Xray (Suppl Table 3)? With such a large chi-squared, how much confidence should reader have in authors' ability to distinguish different magnetic structures? This should be quantified by showing the corresponding difference in chi-squared. Minor remark: what is the meaning of (%) in Supplementary Tables 3 and 4? At least for the reduced chi-squared it does not make sense?

Proper understanding of the stoichiometry and disorder is important because it underlies the central claim of the manuscript,

Line 136-139 > "time-reversal symmetry (TRS) combined with lattice translation ($\equiv \tau_{\vec{Q}}$, see Fig. 2g or 2h), which strictly forbids the finite $\chi_{\vec{Q}}(\omega) = 0$ (and $\chi_{\vec{Q}}(\omega) = 0$ (observed at $T < T_{N2}$ (Fig. 2e-f))."

In general, atomic disorder breaks lattice translation symmetry and therefore its combination with the time-reversal, too. Therefore, strong disorder could probably account for the observed AHE.

There are a number of minor remarks, many of them repeat those in my previous report, but I leave these to the authors' and Editors' discretion.

First, a note on authors' reply to my comments.

>Magnetic orderings, even with novel properties and functionalities, is a narrow subfield of condensed matter physics, so the above claim is an example of grandiose thinking of which there are plenty in the manuscript.

Reply: We respectfully disagree with this statement. Finding materials with novel functionalities have always been the primary motivation of condensed matter physics. Let's review the topics that captured most of the attention of the condensed matter community over the last 40 years: quantum hall systems, high-Tc superconductors, manganites, multiferroics, skyrmion systems, heterostructures, and topological materials. What all these materials have in common is the potential for developing new technologies based on their functionalities.

While my criticism properly describes "Magnetic orderings, even with novel properties and functionalities" as a narrow subfield of condensed matter physics, in their reply authors argue about "materials with novel functionalities", which include quantum hall systems, high-Tc superconductors, multiferroics, non-magnetic topological materials, polymers, semiconductors, etc, etc, - in addition to magnetic orderings. The reply is obviously an example of the red herring fallacy (https://en.wikipedia.org/wiki/Red_herring). In fact, it reflects a systematic deficiency in authors' logics, which is also present in the manuscript. While such logical faults are forgivable in a discussion with a reviewer, they undermine the credibility of the results in a scientific paper.

>We like to remind the referee that the spin and orbital degrees of freedom of an electron can be coupled even in the absence of the relativistic effect.

This reviewer knows well that spin-dependent band splitting of exchange origin is quite common in itinerant electron systems, leading for example to ferromagnetism, or antiferromagnetism. However, the exchange band splitting is distinct from band splitting due to spin-orbit interaction. Calling an exchange splitting of electronic band structure (which describes real-space, "orbital" motion) "effective spin-orbit" is unnecessary, unconventional, and confusing for the reader. None of the references authors cited in their rebuttal calls exchange effects "effective spin-orbit" as authors for some reason insist.

Line 40-41. >one of the primary goals of condensed matter physics -> an important goal of condensed matter physics

Line 49. >The origin of this effective spin-orbital coupling ... -> The origin of this effect ...
Line 82-83 > the tetrahedral triple-Q ordering creates a very strong effective magnetic field of one flux quantum divided by the area of 4 triangular plaquettes in the adiabatic limit.
This is unclear. Previously, authors spoke of an effective exchange field from ordered magnetic moments acting on conduction electrons. This one depends on the ordered moment and on the effective coupling constant. In particular, it changes depending on the spin-species and continuously decreases to zero as ordered moment vanishes with temperature. In other words, it does not quantize. Here therefore, authors mean something else, so they need to explain what it is.

Line 89 > the noncoplanar configuration generates a large effective spin-orbit coupling -> remove altogether, or replace with something like the noncoplanar configuration generates textured exchange field whose effect on the Berry phase is similar to that of spin-orbit coupling.
Authors' fixation on tagging the exchange effects "effective spin-orbit" is difficult to understand, but is also quite misleading. Indeed, it is well known that both spin-orbit interaction and exchange coupling can lead to spin-dependent splitting of the electronic band structure. In the case of exchange interaction, these splittings give rise to eg ferromagnetism, or altermagnetism. This reviewer has not encountered cases where the effects of such exchange splitting of the electronic band structure, which effectively describes real-space, "orbital" motion of electrons, would be called an "effective spin-orbit". Authors' pioneering usage of this onorthodox terminology adds confusion rather than clarity.

Fig 3 caption > a weak line-shaped signal connecting the six M points (= the quadratic mode) -> a weak, diffuse ring-like scattering which we interpret as an evidence of the quadratic mode (predicted by LSWT?)
> line-shaped -> ring-like

Line 111 and on: > Based on the 120° ordering reported in Ref. 25 and a symmetry argument, the authors of Ref. 26 suggested that the observed AHE, $\sigma_{xy}/m(\varphi = 0) \neq 0$, can be attributed to a ferroic order of cluster toroidal dipole moments. However, our latest neutron scattering data reported in this work reveals an entirely different picture: $\text{Co}_{1/3}\text{TaS}_2$ has a magnetic structure with ordering wave vectors of the M-points ($q_m = (1/2, 0, 0)$ and symmetry-related vectors) instead of $q_m = (1/3, 1/3, 0)$.

Line 303-304 >In summary, we have reported a tetrahedral triple-Q ordering in $\text{Co}_{1/3}\text{TaS}_2$, as the only magnetic ground state consistent with our bulk properties and neutron scattering data. The summary above does not sound adequate and needs to be revised. What the manuscript reports is a large amount of experimental data, which is interpreted via invoking "a tetrahedral triple-Q ordering". If manuscript would only report an interpretation as the above summary suggests, it would be of questionable importance and hardly suitable for publication in Nature Communications.

First Report of Referee #1

General Comment: I read the revised draft and replies by the authors. I am happy with the replies and recommend the paper for publication.

Reply: We appreciate the referee's careful consideration and recommendation of our manuscript.

First Report of Referee #2

General Comment: In the revised manuscript authors have addressed large fraction of my comments and suggestions. While the ensuing revisions have notably improved the manuscript, a number of deficiencies still remain, which leave me less than convinced in the validity of the results and interpretations that authors present. Hence, while I maintain that manuscript presents large number of interesting experimental results which are potentially of sufficient importance to warrant publication, I cannot recommend acceptance of the manuscript in its present form.

Reply: We extend our appreciation to the referee for their careful examination of our manuscript. In response to their suggestions, we have taken extra steps to provide further clarification regarding the concerns they raised. Although there are instances where our viewpoints diverge from the comments made in the report, we genuinely value the constructive nature of the feedback. In addition to reviewing our detailed responses addressing each point raised by the reviewer, we kindly encourage them to explore the key references cited within our manuscript. It appears there may be some confusion surrounding the distinction between the 'exchange field' and the effective gauge field, which is generated by the exchange field only in the presence of relativistic spin-orbit coupling and/or non-coplanar ordering (see <https://doi.org/10.1103/PhysRevB.101.024420>).

Comment #1: The most important problem of the manuscript is the discrepancy of the presented magnetic structure refinement with that previously reported in Ref. 25. It is this discrepancy that makes grounds for the new triple-Q scenario compared to the "toroidal" scenario previously "discovered" by some of the authors. It is unclear, what is the reason for the observed different magnetic structures. In their rebuttal letter, authors seem to suggest that authors of Ref 25 simply made a mistake in magnetic structure refinement. In the manuscript, however, authors sweep this issue under the rug and write,

Line 116-118 >This new observation forced us to conduct a more extensive

investigation and come up with a scenario consistent with all the experimental facts. It is obvious that the scenario proposed in the manuscript is not consistent with the experimental facts reported in Ref. 25. Rather than outright discarding the previous work, authors actually have the burden of proof to explain why their results are different and why reader should trust their results and not those of Ref. 25. In fact, judging by the scientific track record of the authors of both papers, this reviewer would put more trust in the refinement of Ref. 25 (Jane Brown's group).

Reply: Thank you for the comment. We have added a new Supplementary Note to elaborate on the discrepancy of the magnetic ground states between Ref. 25 and our work. We also added a few more sentences about this in the main text and referred to our Supplementary Note. Below, we provide more detailed explanations to clarify potential misunderstandings.

On a personal note, one of the authors (Je-Geun Park) had the fortune to work alongside Dr. Jane Brown at the ILL during his many decades-long visits since the 90s. He met her and developed a huge admiration for her carefulness and a scientist as she was, which was one of the reasons why it took us a little bit longer than necessary to come to a more accurate answer.

1) Interpretation of the neutron data of $\text{Co}_{1/3}\text{TaS}_2$.

We wish to emphasize that any misinterpretation of the neutron scattering data can be definitively ruled out. We have chosen to provide this detailed explanation because we are concerned that the referee may have doubts regarding the determination of the ordering wave vector between $\mathbf{q}_m = (1/2, 0, 0)$ (our result) and $\mathbf{q}_m = (1/3, 1/3, 0)$ (ordering wave vector reported in Ref. 25). This concern is based on the following comment:

*“In fact, judging by the scientific track record of the authors of both papers, this reviewer would put more **trust in the refinement** of Ref. 25 (Jane Brown's group).”*

We fully appreciate the referee's consideration, especially given the esteemed reputation of the authors in Ref. 25 and their significant contributions to the neutron scattering field. However, it is crucial to emphasize that the determination of the ordering wave vector as $\mathbf{q}_m = (1/2, 0, 0)$ (our result) as opposed to $\mathbf{q}_m = (1/3, 1/3, 0)$ (as reported in Ref. 25) is not a matter open to interpretation or change. As the referee knows, the positions of $\mathbf{q}_m = (1/2, 0, 0)$ and $\mathbf{q}_m = (1/3, 1/3, 0)$ in the reciprocal space are entirely distinct, even in the momentum magnitude $|\mathbf{Q}|$. Consequently, this distinction can be readily discerned by simply examining the positions of the magnetic Bragg peaks in the data.

Furthermore, it is essential to clarify that while Ref. 25 suggested the ordering wave vector of $\mathbf{q}_m = (1/3, 1/3, 0)$, they did not conduct any refinement to identify a detailed spin configuration based on \mathbf{q}_m . Ultimately, as the reviewer with expertise in the field would agree, one has to trust the data and make an unbiased judgment when evaluating the presented data

rather than placing undue emphasis on the 'scientific track record of the authors of a 40-year-old paper no matter how much respect we have for those scientists.

To aid the referee in gaining a clearer understanding of the situation, we have outlined the following facts based solely on experimental observations, without any additional interpretations:

- ✓ As shown in Fig. 2 (even with the single-crystal data), the magnetic Bragg peaks of our $\text{Co}_{1/3}\text{TaS}_2$ samples appear at the M points ($\mathbf{q}_m = (1/2, 0, 0)$) of the Brillouin zone instead of the K points ($\mathbf{q}_m = (1/3, 1/3, 0)$).
- ✓ We have consistently observed that the $\text{Co}_{1/3}\text{TaS}_2$ samples with two-step phase transitions, large spontaneous Hall conductivity, and weak ferromagnetic moment, manifest $\mathbf{q}_m = (1/2, 0, 0)$. Note that these properties remain unchanged even when slightly tuning the Co composition (see our responses to Comment #2-1).
- ✓ Our conclusion was also confirmed by one recent work on $\text{Co}_{1/3}\text{TaS}_2$ from Tokyo University, published in *Nature Physics* just a few months ago (Takagi *et al.* *Nature Physics* **19**, 961–968 (2023)). Let us make it clear that both groups carried out their work independently until we came across each other's work at the latest APS March meeting this year. We both immediately uploaded two papers: our work at <https://arxiv.org/abs/2303.03760> and UT's work at <https://arxiv.org/abs/2303.04879>. For their case, using single-crystal neutron diffraction, they arrived at precisely the same conclusion as ours. This report further supports the validity of our observations of $\mathbf{q}_m = (1/2, 0, 0)$, rather than the result of Ref. 25. Additionally, it is worth noting that our initial manuscript was submitted in September 2022, well before this work was registered on arXiv and subsequently published in *Nature Physics*. Furthermore, our research is bolstered by an extensive collection of datasets, including ARPES and inelastic neutron scattering, and we also developed a comprehensive theoretical model to elucidate the triple-Q ground state. In contrast, Takagi et al. reached their conclusions based solely on neutron diffraction. Consequently, the originality and novelty of our results and conclusions remain unaffected by their work. We have included a brief note about this related work from Univ. Tokyo at the end of our main text for reference.
- ✓ While Ref. 25 made mention of the observation of 'weak' magnetic peaks corresponding to $\mathbf{q}_m = (1/3, 1/3, 0)$, we were unable to locate any pertinent neutron diffraction data within Ref. 25. Consequently, it has proven challenging for us to access detailed information regarding their findings.
- ✓ Because of the limited information available, it remains unclear whether the sample discussed in Ref. 25 exhibits characteristics such as spontaneous Hall conductivity and a

weak ferromagnetic moment. Notably, there is no mention in their work of observing these specific properties.

As the referee has pointed out, it is indeed reasonable to suspect that there may be differences in sample quality and Co disorder between the sample used in Ref. 25 and our work. We have provided a detailed explanation of this in our response to Comment #2-1.

Comment #2-1: There are also other explanations which question the scenario proposed in the manuscript. Perhaps, the sample used in the present study and that of Ref 25 are somehow different? Perhaps, they differ in stoichiometry and/or degree of structural disorder? Here, I note that authors' own Xray structural refinement indicates lower occupancy of the Co sites (and this is consistent with neutron refinement) and a very large displacement parameter for Co, up to an order of magnitude larger than that for other atoms. Can these imperfections impact the authors' observations and explain their divergence from the previous results of Ref. 25?

Reply: Thank you for your comment. Indeed, the disparities in properties between our sample and the one used in Ref. 25 can be attributed to variations in Co stoichiometry and disorder. The referee's apprehension regarding disorder is valid, and to some extent, it is inevitable, given the nature of intercalation. However, we wish to clarify that we have conducted a thorough investigation into the composition-dependent magnetic properties of $\text{Co}_{1/3}\text{TaS}_2$. As will become evident in our comprehensive response below, the level of disorder in the samples with $\mathbf{q}_m = (1/2, 0, 0)$ is not as significant as the referee may be concerned about. The observations presented in this work can be regarded as intrinsic phenomena rather than being solely attributed to disorder.

1) A possibility of Co composition difference between our sample and that of Ref. 25

Based on our experimental findings and the information available from Ref. 25, it appears that differences in Co stoichiometry and disorder are the most plausible explanations for the variation in ordering wave vectors between our study and Ref. 25. Nevertheless, it's important to note that our case aligns more closely with the ideal stoichiometry, as elaborated upon below.

For Co_xTaS_2 within $0.299(4) < x < 0.325(4)$, the key bulk properties presented in our manuscript – an ordering wave vector of $\mathbf{q}_m = (1/2, 0, 0)$, two successive phase transitions, large spontaneous Hall conductivity, and weak ferromagnetic moment – remain intact. In other words, they can be found in the sample with Co composition quite close to the ideal limit $x = 1/3$ but remain qualitatively the same across a broad Co vacancy concentration of 2.5% ~ 10%. This indicates that the observed properties are not driven by Co vacancies. Fig. R1 demonstrates some of our results, taken after Fig. 2f and Fig. S6 of Ref. 26 (our previous work that reported, for the first time, large spontaneous Hall conductivity in $\text{Co}_{1/3}\text{TaS}_2$).

Notably, $\text{Co}_{1/3}\text{TaS}_2$ samples with slight variations in Co compositions consistently display a similar magnitude of anomalous Hall conductivity ($\sigma_{xy}(H = 0)$) regardless of their longitudinal conductivity (σ_{xx}). This explicitly suggests an intrinsic origin for $\sigma_{xy}(H = 0)$ with little correlation to the specific composition. Consequently, even though the sample we employed for XRD measurements (as indicated in Supplementary Table 1 in the previous version) exhibited occupancies corresponding to $x \sim 0.308$, we want to emphasize that characteristics such as $\mathbf{q}_m = (1/2, 0, 0)$ are still observed in samples with significantly improved Co occupancies. Also, see our reply 3) to Comment #2-1.

The estimated Co composition of the sample used in Ref. 25 can be found in Ref. 24, as indicated by the following statement in Ref. 25: “Crystals of $\text{Co}_{1/3}\text{TaS}_2$ were taken from the same batch as that used by Parkin and Friend (1980a, b).” According to Ref. 24, the composition is approximately $x \sim 0.29$. While we acknowledge that the accuracy of this estimate may be uncertain, it is important to note that it does not align with our investigated range ($0.3 < x < 0.325$). Additionally, this composition is notably further from the ideal stoichiometry ($x = 1/3$) than our sample.

Finally, even after our investigation that covers a broader range of Co composition $0.299 < x < 0.34$ (which includes the ideal limit $x = 1/3$), we could not find any sample exhibiting $\mathbf{q}_m = (1/3, 1/3, 0)$ ordering. Thus, $\mathbf{q}_m = (1/3, 1/3, 0)$ would be outside $0.3 < x < 0.34$. Nevertheless, even when considering these off-stoichiometries, comprehending the emergence of $\mathbf{q}_m = (1/3, 1/3, 0)$ remains challenging, as outlined in our response to Comment #1.

To summarize, our results with $\mathbf{q}_m = (1/2, 0, 0)$ and spontaneous Hall conductivity are found in samples not far from the ideal stoichiometry, so they should be considered intrinsic phenomena. On the other hand, several pieces of circumstantial evidence suggest that $\mathbf{q}_m = (1/3, 1/3, 0)$ in Ref. 25 is not the ordering wave vector of the stoichiometric sample.

Fig. R1 Measured longitudinal and transverse resistivity of Co_xTaS_2 with $0.313(3) < x < 0.325(4)$. The right figure shows their measured anomalous Hall conductivity as a function of longitudinal conductivity. Figures are adapted from Ref. 26.

2) Assessing Co disorder of Ref. 25's sample and our sample

Besides the reported value $x = 0.29$, no additional information is publicly available regarding the structural characterization of the sample used in Ref. 25, such as X-ray or (nuclear) neutron diffraction profiles. Consequently, conducting a more comprehensive comparison beyond composition has been unfeasible. In lieu of this, we have provided an extensive account of our established protocol for evaluating Co disorder in our samples and the criteria we employed to select samples with minimal disorder (some of which are already outlined in the Methods section).

Primarily, we carefully inspected the intercalation profile of Co atoms by assessing the sharpness of the (101) Bragg peak and the Raman peak at 137 cm^{-1} . These peaks correspond to the superlattice peak from Co intercalation and the phonon mode predominantly governed by in-plane vibrations of Co atoms. The measured powder XRD and Raman spectra remained consistent across different x values we investigated, indicating a uniform crystal structure without noticeable emergence of disorder. Also, the full widths at half maximum (FWHMs) of these peaks remain sharp enough to be considered of fine sample quality.

That being said, considering the inherent uncertainties associated with the analysis of vacancies and potential disorder from XRD/Raman profiles, we have concluded that evaluating sample quality through the measurement of the transition temperature ($T_{\text{N}2}$) via magnetometry is a more sensitive and accurate approach. Our investigations have revealed that $T_{\text{N}2}$ gradually decreases as x decreases, and it would exhibit some changes in response to Co disorder beyond vacancies. Unless there is a significant departure from the ideal stoichiometry, $T_{\text{N}2}$ has consistently measured at 26.5 K, a value that stands as the highest we have observed throughout our extensive experience with $\text{Co}_{1/3}\text{TaS}_2$ crystal synthesis, spanning approximately 40 different batches. It is important to note that all measurements presented in this manuscript were conducted exclusively on $\text{Co}_{1/3}\text{TaS}_2$ samples exhibiting $T_{\text{N}2} = 26.5 \text{ K}$. However, we have also verified that qualitatively, the same bulk properties are found in samples with slightly lower $T_{\text{N}2}$ values.

3) Reply to the large Co displacement parameter

Firstly, we acknowledge the typo specifying the refined B_{iso} as $U_{\text{iso}} = B_{\text{iso}}/8\pi^2$. While we believe this error was corrected in the last revision, it appears that it was not adequately taken into account in the submitted version for some reason. We apologize for any confusion this may have caused.

Secondly, we understand the importance of providing accurate and representative data regarding Co displacement parameters, especially in the context of sample quality. We want to draw attention to the fact that much smaller Co displacement parameters (relative to other sites) can be obtained when analyzing the XRD pattern of a sample with a better composition than the one listed in the old Supplementary Table 1. An example of this is already included in Ref. 26, which is our previous work (see Fig. R2 below). The single crystal used for this XRD measurement came from the sample batch where the ordering wave vector of $\mathbf{q}_m = (1/2, 0, 0)$,

two successive phase transitions, weak ferromagnetic moment, and spontaneous Hall conductivity were all clearly confirmed. While we did not refine the occupancy for this dataset, EDX and ICP-AES measurements on this sample batch yielded $x = 0.325(4)$ (corresponding to the batch of sample 4 in Fig. R1). Therefore, although the large B_{iso} values in the old Supplementary Table 1 may indicate some Co disorder for that particular powder sample, it is essential to note that properties like $\mathbf{q}_m = (1/2, 0, 0)$ reported in this paper appear independent of that issue.

In light of the referee's comments, we have taken extra steps to prevent any potential misunderstanding by removing Supplementary Fig. 1 and Supplementary Table 1 in the new version of the manuscript. Instead, we have referred to our previous work (Ref. 26), which contains the XRD measurement results with a more accurate representation of sample composition and quality.

Fig. R2 Refinement of the single-crystal XRD data from the sample with $x = 0.325(4)$. Adapted from Ref. 26 (our previous work). Note that the $2d$ Wychoff site corresponds to Co site disorder. Thus, vanishing occupancy at $2d$ indicates no such disorder in our sample.

Comment #2-2: Additionally, a comparison of the neutron powder diffraction patterns at 3K and 60K shown in Fig. 2a reveals that aside from magnetic reflections, also a couple of structural reflections disappear at 60K: peak at $\sim 1.7 \text{ \AA}^{-1}$, to the right of $(1/2 \ 0 \ 3)$ magnetic reflection, also peaks at ~ 2.3 and $\sim 2.7 \text{ \AA}^{-1}$. If these peaks are not magnetic (and they are not marked as magnetic, but are marked as structural), then the magnetic structure that authors have refined is in doubt. If these peaks are structural indeed, then the transition is magnetostructural in origin and the results of the corresponding structural change need to be understood. This is also corroborated by the first-order type singularity in magnetization (Fig. 1h) at the transition temperature which authors tag as TN2. It is not implausible that some lowering of the magnetostructural symmetry indicated by these peaks might explain the observed AHE in the weakly ferromagnetic phase at low T? Authors need to explain and disentangle these observations.

Reply: It is crucial to distinguish between magnetic and nuclear reflections correctly. The reflections pointed out by the referee are all magnetic reflections, as indicated by the red vertical lines in Fig. R3. There is no change in the nuclear reflections (grey vertical lines) for temperatures below and above the magnetic transition. For instance, the peak at 1.7 \AA^{-1} is $(3/2, -1, 1)$ or its symmetry-equivalent positions. The new caption of Fig. 4 better describes these vertical lines. In summary, there is no discernible change in the nuclear reflections when varying temperatures across the magnetic transition.

Fig. R3 Fig. 4(a) with more explanations about nuclear/magnetic reflections.

Comment #3: Finally, the intercalated dichalcogenide materials family is known to be extremely sensitive to the stoichiometry of the intercalating 3D metal. The difference of ~ 0.01 - 0.02 in the vicinity of $1/3$, ie between 0.31 and 0.33 might drastically change the system's behavior. That authors find this order of difference between their EDX/ICP chemical analysis results and neutron/Xray refinement of the corresponding site occupancies might indicate substantial degree of disorder in the Co position, which in turn might be crucial for the distinct magnetism in the sample authors studied compared to Ref 25. This is also consistent with large displacement parameter found in Xray diffraction.

Reply: Please see our response to comment #2-1. But for the interest of the referee, we shortly summarize them to convey our arguments better:

- ✓ The key features reported in this work were observed up to $x=0.325(4)$, close enough to $x = 1/3$.
- ✓ As described in the previous reply, we investigated the wide range of compositions that includes $x = 1/3$. However, we could not find any sample that exhibits $\mathbf{q}_m = (1/3, 1/3, 0)$.
- ✓ While we cannot confirm its accuracy, the composition of Ref. 25's sample was reported as $x=0.29$.
- ✓ We have a well-established protocol to assess Co disorder and performed measurements with the samples that passed this quality check. Also, the large displacement parameter does not appear for the samples with better compositions, e.g. $x=0.325(4)$. As a result, it is

safe to say that the disorder's impact on our measured properties is not as significant as the referee might be concerned about.

- ✓ Unfortunately, we could not locate any additional information that would provide further insight into the extent of the disorder in Ref. 25's sample. This lack of data makes it challenging to ascertain whether Ref. 25's sample exhibits similar quality to ours.

Comment #4: The related comment is,

On line 161 authors write: "only non-coplanar triple-Q orderings corresponding to equilateral ($|\Delta_v| = |\Delta_{v'}|$, Fig. 2h) or non-equilateral ($|\Delta_v| \neq |\Delta_{v'}|$) tetrahedral configurations are consistent with our Rietveld refinement of neutron diffraction data" Why is the chi-squared of the neutron refinement (Suppl Table 4) so bad, an order-of-magnitude higher than that of Xray (Suppl Table 3)? With such a large chi-squared, how much confidence should reader have in authors' ability to distinguish different magnetic structures? This should be quantified by showing the corresponding difference in chi-squared. Minor remark: what is the meaning of (%) in Supplementary Tables 3 and 4? At least for the reduced chi-squared it does not make sense?

Reply: Thank you for your comment. Firstly, we acknowledge the typo in the chi-square values and have removed the (%) to avoid further confusion. We apologize for any misunderstanding this may have caused.

Secondly, it's important to clarify that Supplementary Tables 1–2 in the previous version pertain to the structure refinement results above T_N , which are irrelevant to the magnetic structure analysis. However, in response to the comment, the discrepancy in χ^2 values between the X-ray and neutron diffraction data can be attributed to the following factors:

- ✓ **Largely different R_{exp} .** R_{exp} can depend on the data condition (e.g. statistics), so caution is needed when comparing the refinement results from different instruments (especially for different sources: neutron and X-ray) only based on $\chi^2 = (R_{\text{wp}} / R_{\text{exp}})^2$. A large χ^2 can, in principle, arise from better data statistics (due to small R_{exp}), which is the reason why the agreement should be assessed based on multiple factors in addition to χ^2 : unweighted (R_p) and weighted residual (R_{wp}). Please refer to "Toby, B. (2006). *R factors in Rietveld analysis: How good is good enough? Powder Diffraction*, 21(1), 67-70. doi:10.1154/1.2179804". The R_p and R_{wp} values, which are more straightforward and thus are considered more important factors of the refinement, are similar to the results in Supplementary Tables 1 and 2. Also, when directly looking into Supplementary Figs. 1 and 4, we do not find any significant disagreement between the data and simulation (to our knowledge, R_p and R_{wp} smaller than 10 % are generally deemed a decent agreement). Furthermore, the resultant structure information between Supplementary Tables 1 and 2 is not much different after all.

Finally, we show how clearly our data can distinguish different magnetic structure models. Fig. R4 below shows the powder diffraction patterns of various magnetic structures and their magnetic R-factors. It is already clear from the figure that inconsistency with the data is significant for the other magnetic structure candidates (\mathbf{V}_{11}^V , \mathbf{V}_{21}^V , \mathbf{V}_{31}^V , and \mathbf{V}_{32}^V). Indeed, these models have much larger R-factors than our refined magnetic structure. Thus, our data definitely contain enough information to distinguish different magnetic structures.

The necessity of using \mathbf{V}_{22}^V and \mathbf{V}_{41}^V together was already described in the Supplementary Note. Based on triple-Q formalism, this solution corresponds to the tetrahedral triple-Q configuration as already described in our manuscript. Note that the same conclusion (using $\alpha\mathbf{V}_{22}^V + \beta\mathbf{V}_{41}^V$) can be found from the neutron diffraction data refinement in *Nature Physics* **19**, 961–968 (2023). We have added Fig. R4 in our new version of Supplementary Information.

Fig. R4. Comparing the powder diffraction patterns of several different magnetic structure models.

Comment #5: Proper understanding of the stoichiometry and disorder is important because it underlies the central claim of the manuscript,

Line 136-139 > "time-reversal symmetry (TRS) combined with lattice translation ($\equiv \tau\mathcal{Q}\hat{\mathbf{r}}$, see Fig. 2g or 2h), which strictly forbids the finite $\langle \tau \rangle_{\mathbf{r}} = 0$ (and $\langle \tau \rangle_{\mathbf{r}} = 0$ (observed at $T < T_{N2}$ (Fig. 2e–f)."

In general, atomic disorder breaks lattice translation symmetry and therefore its combination with the time-reversal, too. Therefore, strong disorder could probably account for the observed AHE.

Reply: Please see our response to Comments # 2-1 and #3.

Comment #6-1: There are a number of minor remarks, many of them repeat those in my previous report, but I leave these to the authors' and Editors' discretion.

First, a note on authors' reply to my comments.

>Magnetic orderings, even with novel properties and functionalities, is a narrow subfield

of condensed matter physics, so the above claim is an example of grandiose thinking of which there are plenty in the manuscript.

Reply: We respectfully disagree with this statement. Finding materials with novel functionalities have always been the primary motivation of condensed matter physics. Let's review the topics that captured most of the attention of the condensed matter community over

the last 40 years: quantum hall systems, high-Tc superconductors, manganites, multiferroics, skyrmion systems, heterostructures, and topological materials. What all these materials have in common is the potential for developing new technologies based on their functionalities.

While my criticism properly describes "Magnetic orderings, even with novel properties and functionalities" as a narrow subfield of condensed matter physics, in their reply authors argue about "materials with novel functionalities", which include quantum hall systems, high-Tc superconductors, multiferroics, non-magnetic topological materials, polymers, semiconductors, etc, etc, - in addition to magnetic orderings. The reply is obviously an example of the red herring fallacy (https://en.wikipedia.org/wiki/Red_herring). In fact, it reflects a systematic deficiency in authors' logics, which is also present in the manuscript. While such logical faults are forgivable in a discussion with a reviewer, they undermine the credibility of the results in a scientific paper.

Reply: Thank you for the comment, and we are sorry for the misunderstanding. We revised the relevant sentence based on the suggestion in Comment #6-3.

Comment #6-2: 8>We like to remind the referee that the spin and orbital degrees of freedom of an electron can be coupled even in the absence of the relativistic effect.

This reviewer knows well that spin-dependent band splitting of exchange origin is quite common in itinerant electron systems, leading for example to ferromagnetism, or altermagnetism. However, the exchange band splitting is distinct from band splitting due to spin-orbit interaction. Calling an exchange splitting of electronic band structure

(which describes real-space, "orbital" motion) "effective spin-orbit" is unnecessary, unconventional, and confusing for the reader. None of the references authors cited in their rebuttal calls exchange effects "effective spin-orbit" as authors for some reason insist.

Reply: We have removed the words "effective spin-orbit" to avoid any potential confusion.

Comment #6-3: Line 40-41. >one of the primary goals of condensed matter physics - > an important goal of condensed matter physics

Reply: We have corrected the sentences accordingly.

Comment #6-4: Line 49. >The origin of this effective spin-orbital coupling ... -> The origin of this effect ...

Reply: We have corrected the sentences accordingly.

Comment #6-5: Line 82-83 > the tetrahedral triple-Q ordering creates a very strong effective magnetic field of one flux quantum divided by the area of 4 triangular plaquettes in the adiabatic limit.

This is unclear. Previously, authors spoke of an effective exchange field from ordered magnetic moments acting on conduction electrons. This one depends on the ordered moment and on the effective coupling constant. In particular, it changes depending on the spin-species and continuously decreases to zero as ordered moment vanishes with temperature. In other words, it does not quantize. Here therefore, authors mean something else, so they need to explain what it is.

Reply: Thank you for the comment. The keywords here are "in the adiabatic limit". In this limit, the spin of the conduction electron remains parallel to the spin of the local moment. Under this adiabatic condition, the wave function of a conduction electron that moves in a loop picks up a Berry phase equal to half of the solid angle spanned by the underlying localized moments enclosed by the loop (see the introduction of the manuscript).

As we explain in the introduction, this Berry phase cannot be distinguished from the Aharonov-Bohm phase induced by the *orbital coupling* to an external magnetic field (this is the field we are referring to). Since any set of three spins of the tetrahedrally ordered phase span $\frac{1}{4}$ of the full solid angle of the sphere, the effective magnetic flux induced by the tetrahedral ordering, in the adiabatic limit, is equal to $\frac{1}{4}$ of a flux quantum of the fictitious magnetic field.

The limit of vanishing ordered moment is opposite to the adiabatic limit because the coupling between the conduction electrons and the localized moment becomes arbitrarily small (the moment of the conduction electron is not aligned anymore with the local magnetic moment

in this regime). It is unclear to us what the referee means by “it does not quantize”. The above explanation indeed applies to the zero-temperature state in the adiabatic limit. Naturally, a finite temperature will induce fluctuations of the solid angle spanned by the spins, and the magnitude of the Hall effect produced by the magnetic ordering will vary as a function of increasing T going to zero at $T=T_N$; see the figure below, which is adapted from Ref. 12.

Comment #6-6: Line 89 > the noncoplanar configuration generates a large effective spin-orbit coupling -> remove altogether, or replace with something like the noncoplanar configuration generates textured exchange field whose effect on the Berry phase is similar to that of spin-orbit coupling.

Authors' fixation on tagging the exchange effects "effective spin-orbit" is difficult to understand, but is also quite misleading. Indeed, it is well known that both spin-orbit interaction and exchange coupling can lead to spin-dependent splitting of the electronic band structure. In the case of exchange interaction, these splittings give rise to eg ferromagnetism, or altermagnetism. This reviewer has not encountered cases where the effects of such exchange splitting of the electronic band structure, which effectively describes real-space, "orbital" motion of electrons, would be called an "effective spin-orbit". Authors' pioneering usage of this onorthodox terminology adds confusion rather than clarity.

Reply: We accept the comment and replaced the sentence with the following one: “the non-coplanar configuration generates an effective gauge field that couples to the orbital degrees of freedom of the conduction electrons.”

Comment #6-7: Fig 3 caption > a weak line-shaped signal connecting the six M points (= the quadratic mode) -> a weak, diffuse ring-like scattering which we interpret as an evidence of the quadratic mode (predicted by LSWT?)
> line-shaped -> ring-like

Reply: We have corrected the sentences accordingly.

Comment #6-8: Line 111 and on: > Based on the 120° ordering reported in Ref. 25 and a symmetry argument, the authors of Ref. 26 suggested that the observed AHE, $\sigma_{xy}/m(\varphi = 0) \neq 0$, can be attributed to a ferroic order of cluster toroidal dipole moments. However, our latest neutron scattering data reported in this work reveals an entirely different picture: $\text{Co}_{1/3}\text{TaS}_2$ has a magnetic structure with ordering wave vectors of the M-points ($q_m = (1/2, 0, 0)$ and symmetry-related vectors) instead of $q_m = (1/3, 1/3, 0)$.

Reply: In response to this comment, we added a few more sentences discussing the differences between our and Ref. 25's results.

Comment #6-9: Line 303-304 >In summary, we have reported a tetrahedral triple-Q ordering in $\text{Co}_{1/3}\text{TaS}_2$, as the only magnetic ground state consistent with our bulk properties and neutron scattering data.

The summary above does not sound adequate and needs to be revised. What the manuscript reports is a large amount of experimental data, which is interpreted via invoking "a tetrahedral triple-Q ordering". If manuscript would only report an interpretation as the above summary suggests, it would be of questionable importance and hardly suitable for publication in Nature Communications.

Reply: Following the referee's suggestion, we have revised the summary to represent the comprehensive contents in our manuscript better. The abstract has also been reinforced accordingly.

Reviewers' Comments:

Reviewer #2:

Remarks to the Author:

In the present revision authors made good faith effort to address comments and suggestions outlined in my previous reports. I would like to commend authors for this endeavor and for taking seriously this reviewer concerns. I particularly commend the detailed and clear explanation of the difference between the magnetic structure reported here and that in the previous reports of Refs. 24, 25. The resultant revisions have made the present manuscript much clearer and easier read - at least for this reviewer, but I believe for the broad readership of Nature Communications as well. I think that the manuscript is suitable for publication in Nature Communicaitons and recommend that it is accepted.

Third Report of Referee #2

General Comment: In the present revision authors made good faith effort to address comments and suggestions outlined in my previous reports. I would like to commend authors for this endeavor and for taking seriously this reviewer concerns. I particularly commend the detailed and clear explanation of the difference between the magnetic structure reported here and that in the previous reports of Refs. 24, 25. The resultant revisions have made the present manuscript much clearer and easier read - at least for this reviewer, but I believe for the broad readership of Nature Communications as well. I think that the manuscript is suitable for publication in Nature Communicaitons and recommend that it is accepted.

Reply: We appreciate the positive evaluation and recommendation of our work for publication.